# Towards a Unified Information-Theoretic Framework for Generalization

**Mahdi Haghifam**[*]
University of Toronto,
Vector Institute

**Gintare Karolina Dziugaite**[†]
Mila

**Shay Moran**
Technion,
Google Research

**Daniel M. Roy**
University of Toronto,
Vector Institute

## Abstract

In this work, we investigate the expressiveness of the "conditional mutual information" (CMI) framework of Steinke and Zakynthinou [1] and the prospect of using it to provide a unified framework for proving generalization bounds in the realizable setting. We first demonstrate that one can use this framework to express non-trivial (but sub-optimal) bounds for any learning algorithm that outputs hypotheses from a class of bounded VC dimension. We then explore two directions of strengthening this bound: (i) Can the CMI framework express optimal bounds for VC classes? (ii) Can the CMI framework be used to analyze algorithms whose output hypothesis space is unrestricted (i.e. has an unbounded VC dimension)?

With respect to Item (i) we prove that the CMI framework yields the optimal bound on the expected risk of *Support Vector Machines* (SVMs) for learning halfspaces. This result is an application of our general result showing that *stable* compression schemes [2] of size $k$ have uniformly bounded CMI of order $O(k)$.

We further show that an inherent limitation of proper learning of VC classes contradicts the existence of a proper learner with constant CMI, and it implies a negative resolution to an open problem of Steinke and Zakynthinou [3]. We further study the CMI of empirical risk minimizers (ERMs) of class $\mathcal{H}$ and show that it is possible to output all consistent classifiers (version space) with bounded CMI *if and only if* $\mathcal{H}$ has a bounded star number [4].

With respect to Item (ii) we prove a general reduction showing that "leave-one-out" analysis is expressible via the CMI framework. As a corollary we investigate the CMI of the one-inclusion-graph algorithm proposed by Haussler et al. [5]. More generally, we show that the CMI framework is universal in the sense that for *every* consistent algorithm and data distribution, the expected risk vanishes as the number of samples diverges *if and only if* its evaluated CMI has sublinear growth with the number of samples.

## 1 Introduction

In this work, we study the expressiveness of generalization bounds in terms of information-theoretic measures of dependence between the output of a learning algorithm and input data. Information-theoretic techniques for proving generalization bounds are powerful; they can provide generalization

---

[*]Part of the work was done the author was an intern at Element AI, a ServiceNow company.
[†]This work was carried out while the author was at ServiceNow. It was finalized at Google Brain.

35th Conference on Neural Information Processing Systems (NeurIPS 2021).

bounds that are algorithm-dependent, data-dependent, and distribution-dependent. This approach was initiated by Russo and Zou [6, 7] and Xu and Raginsky [8] and has since been extended by a number of authors [9–19]. More recently, attention has shifted to whether these techniques can also characterize worst-case (minimax) rates for certain learning problems.

Let $\mathcal{D}$ be an unknown distribution on a space $\mathcal{Z}$, and let $\mathcal{H}$ be a set of classifiers. Consider a (randomized) learning algorithm $\mathcal{A} = (\mathcal{A}_n)_{n \geq 1}$ that selects an element $\hat{h}$ in $\mathcal{H}$, based on $n$ i.i.d. samples $S_n \sim \mathcal{D}^{\otimes n}$, i.e., $\hat{h} = \hat{h}_n = \mathcal{A}_n(S_n)$. The initial focus of this line of work was on the mutual information $I(\mathcal{A}_n(S_n); S_n)$ between the input and the output of a learning problem. This quantity is sometimes referred to as the *input–output mutual information (IOMI)* of an algorithm and denoted by $\mathrm{IOMI}_{\mathcal{D}}(\mathcal{A}_n)$. A natural question is whether the IOMI framework can provide a sharp characterization of the learnability of Vapnik–Chervonenkis (VC) classes, for which we have strong generalization guarantees. A negative resolution was provided by Bassily et al. [12] for the concept class of thresholds in one dimension. Follow up work by Nachum et al. [13] extended the argument in [12], proving the following result:

**Theorem 1.1** (Thm. 1, [13]). *For every $d \in \mathbb{N}$ and every $n \geq 2d^2$, there exists a finite input space $\mathcal{X}$ and a concept class $\mathcal{H} \subseteq \{0,1\}^{\mathcal{X}}$ of VC-dimension $d$ such that, for all proper and consistent learning algorithm $\mathcal{A}_n$, there exists a realizable distribution $\mathcal{D}$ such that $\mathrm{IOMI}_{\mathcal{D}}(\mathcal{A}_n) = \Omega(d \log \log(|\mathcal{X}|/d))$.*

Livni and Moran [18] extended this result even further, showing that, for the class of one-dimensional thresholds over $\{1, \ldots, m\}$, $m \in \mathbb{N}$,[3] for *every* learning algorithm $\mathcal{A}$ there exists a realizable distribution such that either the risk (population loss) is large or the $\mathrm{IOMI}_{\mathcal{D}}(\mathcal{A}_n)$ scales with the cardinality of the space, $m$. These results highlight an important limitation of the IOMI framework: given an unbounded input space, for any "good" learning algorithm there are always scenarios in which $\mathrm{IOMI}_{\mathcal{D}}(\mathcal{A}_n)$ is unbounded. Therefore, the distribution-free learnability of VC classes cannot be expressed using the IOMI framework.

In this paper, we focus on the "*conditional mutual information*" (CMI) framework, proposed by Steinke and Zakynthinou [1]. In order to reason about the generalization error of a learning algorithm, they introduce a super sample that contains the training sample as a random subset and compute the mutual information between the input and output conditional on the super sample (formal definitions are provided in Section 2.1). Improvements of this framework and its application in studying the generalization of specific learning algorithms have been studied in [20–25].

The current paper revolves around the following fundamental question: For which learning problems and learning algorithms is the CMI framework expressive enough to accurately estimate the generalization error? We will focus in particular on whether we can recover optimal worst case (minimax) rates for VC classes satisfying certain properties. The answer to these question provide evidence that the CMI framework provides a unifying framework for studying generalization.

For VC classes, Steinke and Zakynthinou [1] revealed a stark separation between the CMI framework and IOMI framework. They showed the existence of an empirical risk minimization (ERM) algorithm whose CMI is no larger than $d \log n + 2$ for learning every VC class of dimension $d$ given $n$ i.i.d. training samples. In contrast to Theorem 1.1, CMI does not scale with the cardinality of the space. However, the bound on the CMI combined with Steinke and Zakynthinou's CMI-based generalization bound, leads to a bound on the expected excess risk that is suboptimal in some cases by a $\log n$ factor. (For an overview of the known bounds for learning VC classes, please refer to Appendix A.) The suboptimality of their bound prompted Steinke and Zakynthinou [3] to conjecture that the CMI bound for proper learners of VC classes can be improved to $O(d)$. Moreover, Steinke and Zakynthinou connected CMI framework to the sample compression framework of [26] by showing that a sample compression $\mathcal{A}_n$ of size $k$ has $\mathrm{CMI}_{\mathcal{D}}(\mathcal{A}_n) \leq k \log 2n$. Their bound for sample compression schemes is also suboptimal in some cases by a $\log n$ factor.

## 1.1 Contributions

In this paper we extend the reach of the CMI framework by demonstrating its unifying nature for obtaining optimal or near-optimal bounds for the expected excess risk of the various algorithms in the realizable setting.

---

[3]This concept class can be defined as follows. Let $\mathcal{X} = \{1, ..., m\}$. Let $k \in \mathbb{N}$ and $h_k : \mathcal{X} \to \{0, 1\}$ define as $h_k(x) = \mathbb{1}[x > k]$. Then, the class of one-dimensional thresholds over $\{1, ..., m\}$ is $\mathcal{H}_m = \{h_k | k \in \mathbb{N}\}$.

1. We demonstrate that one can use the CMI framework to express non-trivial (but sub-optimal) bounds for every improper learning algorithm that outputs a hypothesis from a class with a bounded VC dimension. This is achieved by an empirical variant of CMI defined by [1].

2. We study the CMI of SVMs for learning half spaces and show that the CMI framework yields optimal bounds on the expected excess risk. Our bound on the CMI of SVM is an application of our general result giving optimal CMI bounds for stable sample compression schemes [2, 27], which improve on CMI bounds for general sample compression schemes [1] by a $\log n$ factor.

3. In the context of proper learning of VC classes, we exhibit VC classes for which the CMI of any proper learner cannot be bounded by any real-valued function of the VC dimension. Then, we consider VC classes with finite star number [4], and prove the existence of a learner with bounded CMI. Finally, we show that the release of the set of all consistent classifiers in $\mathcal{H}$ has bounded CMI *if and only if* $\mathcal{H}$ has finite star number.

4. We show that CMI framwork is *universal* in the realizable setting. More precisely, for every data distribution and consistent learner, the bound on excess risk obtained by the CMI framework vanishes if and only if the excess risk also vanishes as the number training samples diverges. We then show that any learning algorithm with a "leave-one-out" bound of order $O(1/n)$ yields an evaluated-CMI bound of order $O(\log n)$. As an application, we study the classical one-inclusion graph algorithm of Haussler et al. [5] for improper learning of VC classes, and provide a nearly optimal bound on its expected excess risk using the CMI framework. We also prove there exists a randomized one-inclusion graph which learns point functions (singleton) with bounded CMI.

Our results indicate that CMI is a very expressive generalization framework, and one that can tie together existing frameworks. Although most of our results are stated for binary classification in the distribution-free setting, it is interesting to note that the CMI framework is known to provide numerically non-vacuous generalization error guarantees for some modern deep learning models and datasets in the distribution-dependent setting [21, 25]. These developments in a range of different problem settings highlight the importance of understanding the expressiveness of the CMI framework.

## 2  Preliminaries

We consider the problem of binary classification, with inputs in some space $\mathcal{X}$ assigned labels in $\mathcal{Y} = \{0, 1\}$. A concept (or hypothesis) class $\mathcal{H} \subseteq \mathcal{Y}^{\mathcal{X}}$ is a set of functions $h : \mathcal{X} \to \mathcal{Y}$. We say $\mathcal{H}$ *shatters* $(x_1, \ldots, x_m) \in \mathcal{X}^m$ if for all $(y_1, \ldots, y_m) \in \{0, 1\}^m$, there exists $h \in \mathcal{H}$, such that, for all $i \in [m]$, we have $h(x_i) = y_i$. The *VC dimension* of $\mathcal{H}$, denoted by $d$, is the largest $m \in \mathbb{N}$ for which there exists $(x_1, \ldots, x_m) \in \mathcal{X}^m$ shattered by $\mathcal{H}$. If no such finite $m$ exists, then $d = \infty$.

Let $\mathcal{D}$ be a distribution on $\mathcal{Z} = \mathcal{X} \times \mathcal{Y}$. The *empirical (classification) risk* of a classifier $h : \mathcal{X} \to \mathcal{Y}$ on a sample $s = ((x_1, y_1), \ldots, (x_n, y_n)) \in \mathcal{Z}^n$ is $\hat{R}_s(h) = n^{-1} \sum_{i \in [n]} \ell(h, (x_i, y_i))$, where $\ell(h, (x, y)) = \mathbb{1}[h(x) \neq y]$. Let $S_n \sim \mathcal{D}^n$, i.e., let $S_n$ be a sequence of i.i.d. random elements in $\mathcal{Z}$ with common distribution $\mathcal{D}$. (We can view $S_n$ itself as a random element in $\mathcal{Z}^n$.) The *risk* of $h$ is $R_{\mathcal{D}}(h) = \mathbb{E}\hat{R}_{S_n}(h)$, where $\mathbb{E}$ denotes the expectation operator. (The risk has, of course, no dependence on $n$ due to the data being i.i.d.)

A distribution $\mathcal{D}$ is *realizable by a class* $\mathcal{H} \subseteq \mathcal{Y}^{\mathcal{X}}$ if there exists $h \in \mathcal{H}$ such that $R_{\mathcal{D}}(h) = 0$. A sequence $((x_1, y_1), \ldots, (x_n, y_n))$ is said to be *realizable by* $\mathcal{H}$, if for some $h \in \mathcal{H}$, $h(x_i) = y_i$ for all $i \in [n] = \{1, \ldots, n\}$. Note that if a distribution is realizable by $\mathcal{H}$, it implies that with probability one over $S_n \sim \mathcal{D}^n$, the training sample $S_n$ is realizable by $\mathcal{H}$.

Let $\mathcal{A} = (\mathcal{A}_n)_{n \geq 1}$ denote a (potentially randomized) learning algorithm, which, for any positive integer $n$, maps $S_n$ to an element of $\mathcal{X} \to \mathcal{Y}$. We say that $\mathcal{A}$ is a *proper learner for a class* $\mathcal{H} \subseteq \mathcal{X} \to \mathcal{Y}$ if the codomain of $\mathcal{A}_n$ is a subset of $\mathcal{H}$ for every $n$. We say $\mathcal{A}_n$ is a consistent algorithm (learner) if $\hat{R}_{S_n}(\mathcal{A}_n(S_n)) = 0$ a.s. Our primary interest in this paper is the *expected generalization error* of $\mathcal{A}_n$ with respect to $\mathcal{D}$, defined as $\mathrm{EGE}_{\mathcal{D}}(\mathcal{A}_n) = \mathbb{E}[R_{\mathcal{D}}(\mathcal{A}_n(S_n)) - \hat{R}_{S_n}(\mathcal{A}_n(S_n))]$, where we average over both the choice of training sample and the randomness within the algorithm $\mathcal{A}_n$.

### 2.1  Conditional mutual information (CMI) of an algorithm

In order to study generalization, and avoid some of the pitfalls of earlier approaches based on mutual information, Steinke and Zakynthinou [1] propose to study the information contained in a

"supersample" $Z$, a training sample $S_n$ taken from the supersample, and the hypothesis $\mathcal{A}_n(S_n)$ output by a possibly randomized learning algorithm, given $S_n$ as input. Formally, let $Z = (Z_{i,j})_{i \in \{0,1\}, j \in [n]}$ to be an array of i.i.d. random elements in the space $\mathcal{Z}$ of labeled examples, with a common distribution $\mathcal{D}$. In order to choose a training sample $S_n$ of size $n$ from $Z$, let $U = (U_1, U_2, \ldots, U_n)$ be a sequence of i.i.d. Bernoulli random variables in $\{0, 1\}$, independent from $Z$, with $\mathbb{P}(U_i = 0) = 1/2$. Define $S_n = Z_U = (Z_{U_j,j})_{j=1}^n$. The *conditional mutual information (CMI) of* $\mathcal{A}_n$, denoted $\mathrm{CMI}_{\mathcal{D}}(\mathcal{A}_n)$, is defined to be the mutual information between $\mathcal{A}_n(S_n)$ and $U$ given $Z$, denoted $I(\mathcal{A}_n(S_n); U|Z)$. This quantity is equivalent to $I(\mathcal{A}_n(S_n); S_n|Z)$ when $\mathcal{D}$ is atomless, since $(U_1, \ldots, U_n)$ is a.s. measurable with respect to $S_n$ and $Z$. Because $Z$ and $U$ are independent, $\mathrm{CMI}_{\mathcal{D}}(\mathcal{A}_n) \leq \mathrm{H}(U|Z) = \mathrm{H}(U) = n \log 2$. We now pause to introduce this and other information-theoretic quantities formally.

## 2.2 Measures of divergence and information

Let $P, Q$ be probability measures on a measurable space. (We ignore measure-theoretic pathologies for clarity.) For a $P$-integrable or nonnegative function $f$, let $P[f] = \int f \, \mathrm{d}P$. When $Q$ is absolutely continuous with respect to $P$, denoted $Q \ll P$, write $\frac{\mathrm{d}Q}{\mathrm{d}P}$ for (an arbitrary version of) the Radon–Nikodym derivative (or density) of $Q$ with respect to $P$. The *KL divergence* (or *relative entropy*) *of* $Q$ *with respect to* $P$, denoted $\mathrm{KL}(Q \| P)$, is defined as $Q[\log \frac{\mathrm{d}Q}{\mathrm{d}P}]$ when $Q \ll P$ and infinity otherwise.

For a random element $X$ in some measurable space $\mathcal{X}$, let $\mathbb{P}[X]$ denote its distribution, which lives in the space $\mathcal{M}_1(\mathcal{X})$ of all probability measures on $\mathcal{X}$. Given another random element, say $Y$ in $\mathcal{T}$, let $\mathbb{P}^Y[X]$ denote the conditional distribution of $X$ given $Y$. If $X$ and $Y$ are independent, $\mathbb{P}^Y[X] = \mathbb{P}[X]$ a.s. For an event, say $X \in A$, $\mathbb{P}^Y[X \in A]$ denotes the event's conditional probability given $Y$, which is defined to be the conditional expectation of the indicator random variable $\mathbb{1}[X \in A]$ given $Y$, denoted $\mathbb{E}^Y \mathbb{1}[X \in A]$.[4] By the chain (aka tower) rule, $\mathbb{E}\mathbb{E}^{\mathcal{F}} = \mathbb{E}$ for any $\sigma$-algebra $\mathcal{F}$.

The *mutual information between* $X$ *and* $Y$ is $I(X; Y) = \mathrm{KL}(\mathbb{P}[(X, Y)] \| \mathbb{P}[X] \otimes \mathbb{P}[Y])$, where $\otimes$ forms the product measure. Writing $\mathbb{P}^Z[(X, Y)]$ for the conditional distribution of the pair $(X, Y)$ given a random element $Z$, the *disintegrated mutual information between* $X$ *and* $Y$ *given* $Z$, is

$$I^Z(X; Y) = \mathrm{KL}(\mathbb{P}^Z[(X, Y)] \| \mathbb{P}^Z[X] \otimes \mathbb{P}^Z[Y]).$$

Then the *conditional mutual information* of $X$ and $Y$ given $Z$ is $I(X, Y|Z) = \mathbb{E}I^Z(X, Y)$.

Let $\mu = \mathbb{P}[X]$ and let $\kappa(Y) = \mathbb{P}^Y[X]$ a.s. If $X$ concentrates on a countable set $V$ with counting measure $\nu$, the *(Shannon) entropy of* $X$ is $\mathrm{H}(X) = -\mu[\log \frac{\mathrm{d}\mu}{\mathrm{d}\nu}] = -\sum_{x \in V} \mathbb{P}(X = x) \log \mathbb{P}(X = x)$. The *disintegrated entropy of* $X$ *given* $Y$ is defined by $\mathrm{H}^Y(X) = -\kappa(Y)[\log \frac{\mathrm{d}\kappa(Y)}{\mathrm{d}\nu}]$, while the *conditional entropy of* $X$ *given* $Y$ is $\mathrm{H}(X|Y) = \mathbb{E}[\mathrm{H}^Y(X)]$. Note that $\mathrm{H}(X|Y) \leq \mathrm{H}(X)$. We will make use of the following lemma whose proof can be found in [28].

**Lemma 2.1.** *Let* $(X_1, X_2, \ldots, X_n)$ *be a discrete random vector, and* $Y$ *be an arbitrary random variable. Then,* $\mathrm{H}(X_1, \ldots, X_n|Y) \geq \sum_{i=1}^n \mathrm{H}(X_i|X_{-i}, Y)$, *where* $X_{-i} = (X_j : j \in [n], j \neq i)$.

Steinke and Zakynthinou establish a range of generalization bounds in terms of CMI. Our primary interest is in bounds for algorithms that have vanishing empirical risk. For $[0, 1]$-bounded loss, Steinke and Zakynthinou show that

$$\mathbb{E}R_{\mathcal{D}}(\mathcal{A}_n(S_n)) \leq 2\mathbb{E}\hat{R}_S(\mathcal{A}_n(S_n)) + \frac{3\mathrm{CMI}_{\mathcal{D}}(\mathcal{A}_n)}{n}. \tag{1}$$

For consistent learners (i.e., those that achieve zero empirical error a.s.), they also establish

$$\mathbb{E}R_{\mathcal{D}}(\mathcal{A}_n(S_n)) \leq \frac{\mathrm{CMI}_{\mathcal{D}}(\mathcal{A}_n)}{n \log 2}. \tag{2}$$

Steinke and Zakynthinou also introduce a variant of CMI based on the information revealed by the learner's losses on $Z$, rather than by the output hypothesis, $\mathcal{A}_n(S_n)$, directly.

**Definition 2.2** (Evaluated CMI, [1, §6.2.2])**.** Let $L \in \{0, 1\}^{2 \times n}$ be the array with entries $L_{i,j} = \ell(\mathcal{A}_n(S_n), Z_{i,j})$ for $i \in \{0, 1\}$, $j \in [n]$. The *evaluated conditional mutual information of* $\mathcal{A}_n$ *with respect to* $\mathcal{D}$, denoted by $\mathrm{eCMI}_{\mathcal{D}}(\ell(\mathcal{A}_n))$, is the conditional mutual information $I(L; U|Z)$.

---

[4]By definition, $\mathbb{P}^Y[X]$ is a $\sigma(Y)$-measurable random element in $\mathcal{M}_1(\mathcal{X})$, i.e., $\mathbb{P}^Z[U] = \kappa(Z)$ a.s. for some measurable map $\kappa : \mathcal{T} \to \mathcal{M}_1(\mathcal{X})$. More generally, if, say $\mathcal{F} = \sigma(Y, Z)$ is the $\sigma$-algebra generated by $Y$ and $Z$, then a conditional distribution/probability/expectation given $\mathcal{F}$ is a measurable function of $Y$ and $Z$.

By the data processing inequality, $\mathrm{eCMI}_{\mathcal{D}}(\ell(\mathcal{A}_n)) \leq \mathrm{CMI}_{\mathcal{D}}(\mathcal{A}_n)$. Therefore, $\mathrm{eCMI}_{\mathcal{D}}(\ell(\mathcal{A}_n))$ is also bounded above by $n \log 2$. For consistent learners, Steinke and Zakynthinou show

$$\mathbb{E}R_{\mathcal{D}}(\mathcal{A}_n(S_n)) \leq 1.5 \frac{\mathrm{eCMI}_{\mathcal{D}}(\ell(\mathcal{A}_n))}{n}. \tag{3}$$

For consistent learners $\mathcal{A}_n$ with bounded CMI or eCMI, these results imply their expected excess risk is of order $O(1/n)$. The following result gives a nearly optimal bound for the generalization error for VC classes in term of the evaluated CMI. The proof (Appendix B) uses standard arguments, controlling the cardinality of the support of $L$ using the Sauer–Shelah lemma.

**Theorem 2.3.** *For every $n$, let $\mathcal{A}_n : \mathcal{Z}^n \to \mathcal{H}_n$, where $\mathcal{H}_n$ is a concept class with VC dimension $d_n$. Then, for every $n$ and distribution on $Z$, $I^Z(L;U) \leq d_n \log 6n$ a.s. In particular, $\sup_{\mathcal{D}} \mathrm{eCMI}_{\mathcal{D}}(\ell(\mathcal{A}_n)) \in O(d_n \log n)$.*

*Remark* 2.4. Markov's inequality and Eq. (2) imply $\mathbb{P}(R_{\mathcal{D}}(\mathcal{A}_n(S_n)) \geq \epsilon) \leq \mathrm{CMI}_{\mathcal{D}}(\mathcal{A}_n)/(\log(2)n\epsilon)$ for consistent learners. By [21, Thm. 2.1], $I(\mathcal{A}_n(S_n); U|Z) \leq I(\mathcal{A}_n(S_n); S_n)$. This observation, combined with [12, Prop. 11], implies there is an input space, data distribution, and consistent learning algorithm for which this tail bound's dependence on $n$ is *tight*. If one were to obtain sample complexity bounds via such tail bounds, one would only prove that $O(1/(\epsilon\delta))$ samples suffice to find a hypothesis with $\epsilon$ estimation error with probability at least $1 - \delta$. The linear dependence on $1/\delta$ is, however, suboptimal. As such, it seems that the CMI framework cannot be used to obtain optimal sample complexity bounds in the PAC framework. Recent proposals for disintegrated notions of CMI in [22] might provide a framework for studying the sample complexity of PAC learning using an information-theoretic framework. ◁

# 3 Optimal CMI Bound for SVM and Stable Compression Schemes

In this section, we show that the CMI framework can be used to derive an optimal excess risk bound for the SVM algorithm learning half spaces in $\mathbb{R}^d$. To show this, we establish optimal CMI bounds for the subclass of *stable* sample compression schemes, which imply this section's main result:

**Theorem 3.1.** *Let $\mathcal{A}_n$ be the SVM algorithm for learning the class of half spaces in $\mathbb{R}^d$. Then, for every $n > d/2$ and realizable distribution $\mathcal{D}$ in $\mathbb{R}^d$, we have $\mathrm{CMI}_{\mathcal{D}}(\mathcal{A}_n) \leq 2(d + 1) \log 2$.*

Combining this result with Eq. (2) gives $\mathbb{E}R_{\mathcal{D}}(\mathcal{A}_n(S_n)) \leq 2(d+1)/n$. The lower bound for expected excess risk of linear classifiers in [29] shows this bound is optimal up to a constant factor.

## 3.1 CMI of Stable Compression Schemes

Littlestone and Warmuth [26] introduced compression schemes, which capture the idea that a consistent hypothesis can be defined in terms of a fixed number of samples. Formally, for a concept class $\mathcal{H} \subseteq \mathcal{Y}^{\mathcal{X}}$, a *sample compression scheme* of size $k \in \mathbb{N}$ is a pair $(\kappa, \rho)$ of maps such that, for all samples $s = ((x_i, y_i))_{i=1}^n$ of size $n \geq k$, the map $\kappa$ compresses the sample into a length-$k$ subsequence $\kappa(s) \subseteq s$ which the map $\rho$ uses to reconstruct an empirical risk minimizer $\hat{h} = \rho(\kappa(s))$. Steinke and Zakynthinou prove the following upper bound on the CMI of a sample compression scheme.

**Theorem 3.2** ([1, Thm. 4.1]). *Let $\mathcal{H}$ be a hypothesis class that has a sample compression scheme $(\kappa, \rho)$ of size $k$. Then, $\mathrm{CMI}_{\mathcal{D}}(\mathcal{A}_n) \leq k \log(2n)$ where $\mathcal{A}_n(\cdot) = \rho(\kappa(\cdot))$.*

Note that the bound in Theorem 3.2 *cannot* be improved from $O(k \log n)$ to $O(k)$ for *every* sample compression scheme, and so the bound in Theorem 3.2 is tight, and cannot be improved without further information about the compression scheme. The proof of the optimally stems from the fact that there exists compression schemes of size $k$ and data distributions $\mathcal{D}$ such that there is a lower bound $\mathbb{E}[R_{\mathcal{D}}(\mathcal{A}_n)] = \Omega(k \log(n)/n)$ where $\mathcal{A}_n(\cdot) = \rho(\kappa(\cdot))$ [30, 31]. Combining this lower bound with Eq. (2) proves the optimally of Theorem 3.2.

Nevertheless, we can circumvent this lower bound by considering an important subclass of the sample compression schemes. Many natural compression schemes are also *stable* in the sense that removing any training example that was not in the compressed sequence does not alter the resulting classifier. To give a formal definition, we write $s \subseteq s'$ for two sequences $s, s'$ if, under some permutation, $s$ is a subsequence of $s'$.

**Definition 3.3** (Stable sample compression scheme; [2]). A sample compression scheme $(\kappa, \rho)$ of size $k$ is said to be *stable* if $\kappa$ is symmetric (i.e., invariant to permutation of its input) and, for every realizable sample $s$ of size $n \geq k$, and every sequence $s'$ such that $\kappa(s) \subseteq s' \subseteq s$, we have $\rho(\kappa(s)) = \rho(\kappa(s'))$. Due to the symmetry of $\kappa$, we refer to its output as the compression *set*, although the equivalence class of sequence under permutations is the structure of a multiset, not a set.

The concept of a stable compression scheme has its roots in the analysis of the SVM for learning half-spaces in $\mathbb{R}^d$ [32], which is the quintessential example of a stable sample compression scheme. For SVMs, the compression (multi)set contains at most $d + 1$ distinct "support vectors" for any given training set. The reconstruction map outputs the max-margin classifier over the set of support vectors. By stability, removing any training example that is not a support vector does not change the resulting classifier [33, Sec. 5.3.2]. In the next theorem, we present a uniform CMI bound over realizable distributions for every stable sample compression scheme. Our bound removes the $\log n$ factor from Theorem 3.2 and is *optimal* up to a constant factor in the distribution-free setting.

**Theorem 3.4.** *Let $\mathcal{H}$ be a concept class with a stable compression scheme $(\kappa, \rho)$ of size $k$. Then, for every realizable data distribution $\mathcal{D}$ and $n \geq k$, $\mathrm{CMI}_{\mathcal{D}}(\mathcal{A}_n) \leq 2k \log 2$, where $\mathcal{A}_n = \rho(\kappa(\cdot))$.*

*Remark* 3.5. Steinke and Zakynthinou [1, Sec. 4.4] propose an algorithm for learning threshold functions (positive rays) in the realizable setting over $\mathbb{R}$ that achieves $\mathrm{CMI}_{\mathcal{D}}(\mathcal{A}_n) \leq 2 \log 2$. It is interesting to note that their algorithm can be viewed as a stable compression scheme. Specifically, for a realizable training set $s$, let $x^\star = \min\{x \in \mathbb{R} : (x, 1) \in s\}$ if $s$ has any sample with label 1, otherwise let $x^\star = \infty$. Then the algorithm proposed by Steinke and Zakynthinou is $\mathcal{A}_n(S_n) = \hat{h}$, where $\hat{h}(x) = \mathbb{1}[x \geq x^\star]$. Steinke and Zakynthinou present a bespoke analysis of this special algorithm. It is straightforward to see that the algorithm is a stable compression scheme of size one and the compression map here is symmetric. Therefore, the result of Theorem 3.4 gives $\mathrm{CMI}_{\mathcal{D}}(\mathcal{A}_n) \leq 2 \log 2$. ◁

*Proof of Theorem 3.4.* Let $W = \mathcal{A}_n(Z_U) = \rho(\kappa(Z_U))$ and note that $\mathcal{A}_n$ is deterministic. We have $\mathrm{H}(U|Z) = n \log 2$ due to independence of $U$ and $Z$ and the independence of components of $U$. Then, by the definition of mutual information in terms of entropy, and Lemma 2.1,

$$\mathrm{CMI}_{\mathcal{D}}(\mathcal{A}_n) = \mathrm{H}(U|Z) - \mathrm{H}(U|W, Z) \leq n \log 2 - \sum_{i=1}^{n} \mathrm{H}(U_i|W, Z, U_{-i}). \tag{4}$$

Fix $i \in [n]$, and define $U_{i \to b} \triangleq (U_1, \ldots, U_{i-1}, b, U_{i+1}, \ldots, U_n)$ for $b \in \{0, 1\}$. Using this notation, we can define two training sets $S_{i \to b} = Z_{U_{i \to b}}$ for $b \in \{0, 1\}$. Let $\mathcal{F}_i$ be the $\sigma$-algebra $\sigma(W, Z, U_{-i})$ and let $E$ be the event $\rho(\kappa(S_{i \to 0})) = \rho(\kappa(S_{i \to 1}))$. Then, by the non-negativity of entropy,

$$\mathrm{H}(U_i|W, Z, U_{-i}) = \mathbb{E}\big[\mathrm{H}^{\mathcal{F}_i}(U_i)\big] \geq \mathbb{E}\big[\mathrm{H}^{\mathcal{F}_i}(U_i)\mathbb{1}[E]\big]. \tag{5}$$

Note that, conditional on the sub-$\sigma$-algebra $\mathcal{G}_i = \sigma(Z, U_{-i})$, $W$ takes on at most two values. However, on the event $E$ (or equivalently, conditioning further on the event $E$, since $E$ is $\mathcal{G}_i$-measurable), $W$ is now nonrandom because it takes on a single value. It follows that, conditional on $\mathcal{G}_i$ and the event $E$, $W$ is trivially independent of every random variable, including $U_i$. Ergo, on $E$, $\mathbb{E}^{\mathcal{G}_i}[U_i] = \mathbb{E}^{\mathcal{F}_i}[U_i] = \mathbb{P}^{\mathcal{F}_i}[U_i = 1]$. But $U_i$ is independent of $\mathcal{G}_i$, and so $\mathbb{E}^{\mathcal{G}_i}[U_i] = \mathbb{E}[U_i] = \frac{1}{2}$. Thus, on $E$, $\mathbb{P}^{\mathcal{F}_i}[U_i = 1] = \frac{1}{2}$ and so $\mathrm{H}^{\mathcal{F}_i}(U_i) = \mathrm{H}_b(\frac{1}{2}) = \log 2$. Therefore,

$$\mathrm{H}(U_i|W, Z, U_{-i}) \geq \log 2 \cdot \mathbb{P}\big(\rho(\kappa(S_{i \to 0})) = \rho(\kappa(S_{i \to 1}))\big). \tag{6}$$

We can bound the probability of $E$ from below using the stability property of the compression scheme. For any $(x_1, \tilde{x}_1, x_2, \ldots, x_n) \in \mathcal{X}^{n+1}$ and $h \in \mathcal{H}$, consider two multisets $S = \{(x_1, h(x_1)), (x_2, h(x_2)), \ldots, (x_n, h(x_n))\}$ and $\tilde{S} = \{(\tilde{x}_1, h(\tilde{x}_1)), (x_2, h(x_2)), \ldots, (x_n, h(x_n))\}$, where $S$ and $\tilde{S}$ differ only in the first element. Define the multiset $S \cup \tilde{S} = \big\{(x_1, h(x_1)), (\tilde{x}_1, h(\tilde{x}_1)), (x_2, h(x_2)), \ldots, (x_n, h(x_n))\big\}$. We claim that if $(x_1, h(x_1))$ and $(\tilde{x}_1, h(\tilde{x}_1))$ are not the members of the compression set $S \cup \tilde{S}$, then $(x_1, h(x_1))$ and $(\tilde{x}_1, h(\tilde{x}_1))$ are not in the compression set of $S$ and $\tilde{S}$, respectively. To prove this claim, since $(x_1, h(x_1))$ is not in the compression set $S \cup \tilde{S}$ by the stability of $\kappa$, we have $\rho(\kappa(S \cup \tilde{S})) = \rho(\kappa(S \cup \tilde{S} \setminus \{(x_1, h(x_1))\}))$. By the definition of $S$ and $\tilde{S}$, $S \cup \tilde{S} \setminus \{(x_1, h(x_1))\} = \tilde{S}$. Thus, combining facts that $(x_1, h(x_1))$ is not in the compression set $S \cup \tilde{S}$ and $\kappa(S \cup \tilde{S}) = \kappa(S \cup \tilde{S} \setminus \{(x_1, h(x_1))\}) = \kappa(\tilde{S})$, we obtain $(\tilde{x}_1, h(\tilde{x}_1))$ is not in the

compression set $\tilde{S}$. Similarly, we can prove $(x_1, h(x_1))$ is not a member of the compression set $S$ by switching $x_1$ with $\tilde{x}_1$ in the argument. By this argument,

$$\mathbb{P}\big(\rho(\kappa(S_{i\to 0})) = \rho(\kappa(S_{i\to 1}))\big) \geq \mathbb{P}\big(Z_{0,i} \notin \kappa(S_{i\to 0} \cup S_{i\to 1}) \wedge Z_{1,i} \notin \kappa(S_{i\to 0} \cup S_{i\to 1})\big). \quad (7)$$

Recall that the elements of $Z$ are i.i.d., hence exchangeable. Since the size of the sample compression is $k$ and $\kappa$ is symmetric, we have

$$\mathbb{P}\big(Z_{0,i} \notin \kappa(S_{i\to 0} \cup S_{i\to 1}) \wedge Z_{1,i} \notin \kappa(S_{i\to 0} \cup S_{i\to 1})\big) \geq \binom{n-1}{k}/\binom{n+1}{k}. \quad (8)$$

Combining Eqs. (6) to (8) yields $\mathrm{H}(U_i|W, Z, U_{-i}) \geq \log 2 \cdot \binom{n-1}{k}/\binom{n+1}{k} \geq (1 - 2k/n)\log 2$. Finally, the result follows by substitution of this bound into Eq. (4). $\qquad\square$

The result for SVMs (Theorem 3.1) follows immediately from Theorem 3.4 and the fact that the SVM may be expressed as a stable compression scheme of size $d + 1$.

# 4 CMI of Proper Learning of VC classes

Following their paper introducing CMI, Steinke and Zakynthinou posed several open problems asking whether VC classes under realizibility admit learners with bounded CMI. We will restate their conjectures, and then showing that there exist some VC classes for which it is not possible to find a proper learner with bounded CMI under realiziblity. We then consider a subset of VC classes, namely VC classes with finite star number, and show that for such concept classes, there exists an ERM with bounded CMI.

We first state the main result of [1] on the CMI of proper learners.

**Theorem 4.1** (Thm. 4.12, [1] ). *Let $\mathcal{H}$ be a concept class with VC dimension $d$. Then for all $n \in \mathbb{N}$, there exists a proper ERM algorithm $\mathcal{A}_n$ for learning $\mathcal{H}$ such that for every realizable distribution $\mathcal{D}$, $\mathrm{CMI}_{\mathcal{D}}(\mathcal{A}_n) = O(d \log n)$.*

*Remark* 4.2 (Comparison of Theorem 4.1 and Theorem 2.3). First, note that Theorem 4.1 does not hold for every ERM algorithm. As discussed in [1], we can construct pathological ERMs with nearly maximal CMI by simply encoding the information $U$ into the "lower-order" bits of $W$.

It is also worth noting that our result in Theorem 2.3 is more general. There we show that a bound $O(d \log n)$ holds for *evaluated* CMI of *any algorithm* that outputs a hypothesis from VC class, whereas Theorem 4.1 holds for a *specific* proper algorithm. $\qquad\triangleleft$

## 4.1 A Limitation of Proper Learning

Steinke and Zakynthinou [3] propose two conjectures regarding CMI for proper learning of VC classes under the realizability assumption, both of which can be seen as special cases of the following statement:

**Statement 1.** *There exists a real-valued function $f$ and constant $c \geq 0$ such that, for every nonnegative integer $d$ and VC class $\mathcal{H} \subseteq \mathcal{X} \to \mathcal{Y}$ of dimension $d$, there exists a proper learning algorithm $\mathcal{A}$ for $\mathcal{H}$ such that, for every $n \geq d$, $\mathrm{CMI}_{\mathcal{D}}(\mathcal{A}_n) \leq f(d)$ for all $\mathcal{D}$ and, for every realizable $s \in \mathcal{Z}^n$, $\mathbb{E}\hat{R}_s(\mathcal{A}_n(s)) \leq c\,d/n$, where the expectation is taken only over the randomness in $\mathcal{A}_n$.*

Steinke and Zakynthinou [3] conjecture that Statement 1 holds for $f$ linear. In this section, we show that Statement 1 is false in general: it is not possible to find a proper learning algorithm for *every* VC class that removes the $\log(n)$ factor from Theorem 4.1. For a class $\mathcal{H} \subseteq \mathcal{X} \to \mathcal{Y}$, let $\mathcal{M}^{\mathcal{H}}_{\mathrm{prop}}(\epsilon, \delta)$ denote the *proper optimal sample complexity* of $(\epsilon, \delta)$-PAC learning $\mathcal{H}$, i.e., $\mathcal{M}^{\mathcal{H}}_{\mathrm{prop}}(\epsilon, \delta)$ is the least integer $n$, for which there exists a proper learning algorithm $\mathcal{A}$ such that, for every realizable distribution $\mathcal{D}$, $\mathbb{P}(R_{\mathcal{D}}(\mathcal{A}_n(S_n)) \geq \epsilon) \leq \delta$. The following result provides a lower-bound on the sample complexity of proper learning:

**Theorem 4.3** (Thm. 11, [2]). *Let $\epsilon \in (0, 1/8)$ and $\delta \in (0, 1/100)$. There exists a concept class with VC dimension $d$. for which we have $\mathcal{M}^{\mathcal{H}}_{prop}(\epsilon, \delta) \geq \frac{\tilde{c}}{\epsilon}(d\,\mathrm{Log}\frac{1}{\epsilon} + \mathrm{Log}\frac{1}{\delta})$ for a fixed numerical constant $\tilde{c} > 0$, where $\mathrm{Log}(x) = \max\{1, \log(x)\}$ for $x \geq 0$.*

We now present the main result: for VC classes, we show that the existence of a learning algorithm with bounded CMI contradicts the lower bound on the sample complexity in Theorem 4.3. The proof can be found in Appendix C.

**Theorem 4.4.** *Statement 1 is false.*

*Remark* 4.5. Consider a modified Statement 1, seeking a proper learner with bounded eCMI instead. We can show that this modified statement is also false. ◁

## 4.2 VC Classes with Finite Star Number

Theorem 4.4 states that it is not possible to find a proper learning algorithm with bounded CMI for *every* VC class. Note that this limitation does not imply a failure of the CMI framework for characterizing the expected excess risk of learning VC classes. Instead, the impossibility can be attributed to an inherent limitation of proper learning algorithms, since there exist VC classes such that no proper learning algorithm $\mathcal{A}_n$ satisfies $\mathbb{E}[R_{\mathcal{D}}(\mathcal{A}_n)] = O(1/n)$ [2]. In this section, we consider a family of VC classes for which we *can* show the existence of a learner with bounded CMI. We begin with some definitions.

Two sequences $((x_1, y_1), \ldots, (x_n, y_n))$ and $((x_1', y_1'), \ldots, (x_n', y_n'))$ are *neighbours* if $x_i = x_i'$ for all $i \in [n]$, and $y_i = y_i'$ for all but exactly one $i \in [n]$. Fix any concept class $\mathcal{H} \subseteq \mathcal{Y}^{\mathcal{X}}$. *Star number of $\mathcal{H}$* [4, Def. 2], denoted by $\mathfrak{s}$, is the largest integer $n$ such that there exists a realizable $s \in (\mathcal{X} \times \mathcal{Y})^n$, and every neighbour of $s$ is realizable by $\mathcal{H}$. If no such largest integer $n$ exists, then $\mathfrak{s} = \infty$. Hanneke and Yang [4, Sec. 4.1] calculate the star number of some common concept classes. It is straightforward to see that $d \leq \mathfrak{s}$. For any $n \in \mathbb{N}$, and $s = ((x_1, y_1), \ldots, (x_n, y_n)) \in (\mathcal{X} \times \mathcal{Y})^n$, define a *version space* of $s$ with respect to $\mathcal{H}$ as $V_{\mathcal{H}}[s] = \{h \in \mathcal{H} : \hat{R}_s(h) = 0\}$, a set of classifiers that are consistent with $s$.

### 4.2.1 Star Number, Version Space, and CMI

Fix any concept class $\mathcal{H}$, and assume that, after observing a training sample $S_n$, we want to output the version space $V_{\mathcal{H}}[S_n]$, i.e., the set of all classifiers consistent with $S_n$. We are interested in the following question: for which concept classes does the version space carry little information about the training samples conditioned on the supersample? More precisely, for which classes is $I(V_{\mathcal{H}}[S_n]; U|Z) = O(1)$? Note that bounding the "CMI" of the version space provides a bound on the CMI of a broad class of algorithms that choose a particular ERM based solely on the version space, potentially under further constraints, such as privacy, fairness, etc.

In this section, we give a complete characterization of when $I(V_{\mathcal{H}}[S_n]; U|Z) = O(1)$, and show that it is possible if and only if $\mathcal{H}$ has finite star number. In particular, given a class with infinite star number, we demonstrate that $I(V_{\mathcal{H}}[S_n]; U|Z) = \Omega(n)$. We begin with an upper bound, whose proof can be found in Appendix D.

**Theorem 4.6.** *Let $n \in \mathbb{N}$, $\mathcal{H}$ be a concept class with star number $\mathfrak{s}$, and $\mathcal{D}$ be a realizable distribution. Let $Z$, $U$, and $S_n$ be as defined in the beginning of this section. Then for every $n \geq \mathfrak{s}$, we have $I(V_{\mathcal{H}}[S_n]; U|Z) \leq 2\mathfrak{s} \log 2$.*

We can use the data processing inequality and Theorem 4.6 to obtain the following:

**Corollary 4.7.** *Let $\mathcal{H}$ be a concept class with the star number $\mathfrak{s}$. Consider any ERM algorithm $\mathcal{A}_n$ for which the Markov chain $S_n - V_{\mathcal{H}}[S_n] - \mathcal{A}_n(S_n)$ holds; in other words, the output of the algorithm and the training set are conditionally independent given the version space. Then, for any such an algorithm, for every $n \geq \mathfrak{s}$, and every realizable distribution $\mathcal{D}$, we have $\mathrm{CMI}_{\mathcal{D}}(\mathcal{A}_n) \leq 2\mathfrak{s} \log 2$.*

In Corollary 4.7, by assuming the Markov structure $S_n - V_{\mathcal{H}}[S_n] - \mathcal{A}_n(S_n)$ we restrict the information of the ERM algorithm $\mathcal{A}_n(S_n)$. One might try to extend our result in Corollary 4.7 such that it holds for *any* ERM without any constraints. However, for the class of one-dimensional threshold over $\mathbb{R}$, whose star number is two, one can construct an ERM with maximal CMI [1, Sec. 4.3]. Therefore, the Markov chain assumption cannot be removed. The next theorem shows $\mathfrak{s} < \infty$ is a necessary condition, for otherwise, there exist learning scenarios under which we cannot output the version space, even with merely sublinear CMI.

**Theorem 4.8.** *For every $n \in \mathbb{N}$, $n \geq 2$ and for every concept class $\mathcal{H}$ with star number $\mathfrak{s}$ with $\mathfrak{s} \geq 2$ over input space $\mathcal{X}$, there exists a realizable data distribution $\mathcal{D}$ on $\mathcal{X} \times \mathcal{Y}$ such that $I(V_{\mathcal{H}}[S_n]; U|Z) = \Omega(\min\{\mathfrak{s}, n\})$.*

*Proof sketch:* Let $\mathcal{X} = [n]$ and consider the concept class $\mathcal{H} = \{h_0, h_1, \ldots, h_n : \mathcal{X} \to \mathcal{Y}\}$, where $h_0(x) = 0$ is the zero function and $h_t(x) = \mathbb{1}[x = t]$, for $t \in [n]$, are point functions. It is easy to

see that this concept class has star number $n$ on $\mathcal{X}$. Let $\mathcal{D}$ correspond to the uniform distribution on $\mathcal{X}$ and target function $h_0$. Consider the bijection between $\mathcal{H}$ and $\{0, 1, \ldots, n\} \supseteq \mathcal{X}$. For every training sequence, the version space contains 0 and every point in $\mathcal{X}$ not observed in $S_n = Z_U$. The key observation is that, in each column of $Z$, one point was *not* selected for training, and so each column contains zero or one points in the version space. Whenever there is one point, the value of $U_i$ is revealed for that column. We show that the number of columns with this property is a lower bound on $I(\mathrm{V}_{\mathcal{H}}[S_n]; U|Z)$. A coupon collector's argument yields a lower bound the number of such columns. The formal proof can be found in Appendix E. □

### 4.2.2 An ERM whose CMI is logarithmic in star number

In the next theorem, we show that there exists an ERM for learning VC classes with a finite star number for which the CMI is upper bounded by a constant and its dependence on star number is logarithmic. The proof is provided in Appendix F.

**Theorem 4.9.** *Let $\mathcal{H}$ be a concept class with VC dimension $d$ and star number $\mathfrak{s}$. Then, there exists an ERM $\mathcal{A}_n$ for learning $\mathcal{H}$ such that for every $n \geq \mathfrak{s}$ and for every realizable distribution $\mathcal{D}$, we have $\mathrm{CMI}_{\mathcal{D}}(\mathcal{A}_n) = O\big(d\log(\mathfrak{s}/d)\big)$.*

Note that Theorem 4.9 shows the existence of a specific ERM with constant CMI, whereas in Corollary 4.7 we show a broad class of ERMs has bounded CMI.

## 5 Universality of eCMI and Improper Learning of VC Classes

The eCMI, introduced in Definition 2.2, is an appropriate information-theoretic notion for analyzing learning algorithms when there is no natural parameterization of the set of possible predictors, such as for improper or transductive algorithms. In this section, we show that eCMI is *universal* in the realizable setting. Then, we show that the CMI framework can be used to obtain a near-optimal bound on the expected excess risk of any algorithm with a leave-one-out error guarantee. As an application, we study CMI of the classical *one-inclusion graph prediction* algorithm, which was first proposed by Haussler et al. [5] as an optimal improper learner for VC classes. The next theorem is the main result of this section, whose proof can be found in Appendix G.

**Theorem 5.1.** *Let $n \geq 2 \in \mathbb{N}$, let $\mathcal{A}_n$ be a learning algorithm, and let $\mathcal{D}$ be a distribution on $\mathcal{Z}$. Assume with probability one $\hat{R}_{S_n}(\mathcal{A}_n(S_n)) = 0$. Then,*

$$2/3 R_{\mathcal{D}}(\mathcal{A}_n) \overset{(a)}{\leq} \mathrm{eCMI}_{\mathcal{D}}(\ell(\mathcal{A}_n))/n \overset{(b)}{\leq} \mathrm{H}_b(R_{\mathcal{D}}(\mathcal{A}_n)) + R_{\mathcal{D}}(\mathcal{A}_n)\log(2), \qquad (9)$$

*where $\mathrm{H}_b(\cdot)$ is the binary entropy function, and $R_{\mathcal{D}}(\mathcal{A}_n) = \mathbb{E}[R_{\mathcal{D}}(\mathcal{A}_n(S_n))]$.*

The inequality $(a)$ in Eq. (9) implies that, if $\mathrm{eCMI}_{\mathcal{D}}(\ell(\mathcal{A}_n))/n$ vanishes as $n$ diver2ges, then $R_{\mathcal{D}}(\mathcal{A}_n)$ vanishes as well. The inequality $(b)$ is more interesting: it implies that, if $R_{\mathcal{D}}(\mathcal{A}_n)$ vanishes as $n$ diverges, then $\mathrm{eCMI}_{\mathcal{D}}(\ell(\mathcal{A}_n))/n$ also vanishes.

Assume that a consistent algorithm $\mathcal{A}$ satisfies $R_{\mathcal{D}}(\mathcal{A}_n) = \theta/n$ for $\theta \in \mathbb{R} \geq 1$. Then, it is straightforward to see from Direction $(b)$ in Eq. (9) that $\mathrm{eCMI}_{\mathcal{D}}(\ell(\mathcal{A}_n))/n = O(\theta\log(n))$. Also, for an algorithm with $R_{\mathcal{D}}(\mathcal{A}_n) = \theta\log(n)/n$ the upper bound in Eq. (9) is given by $O(\theta(\log(n))^2)$. This observation suggests that our upper bound for $\mathrm{eCMI}_{\mathcal{D}}(\ell(\mathcal{A}_n))/n$ in Eq. (9) provides a bound on the expected excess risk which is sub-optimal by a $\log(n)$ factor in some interesting cases.

*Remark* 5.2. Note that the result in Theorem 5.1 *does not* imply our results in former sections. In particular our results in Theorem 2.3, Theorem 3.4, Corollary 4.7, and (later in) Theorem 5.6 show that CMI framework provides *optimal* characterization of the expected excess risk in the considered scenarios. ◁

The following corollary summarizes our result for the consistent algorithms with a leave-one-out error guarantee.

**Corollary 5.3.** *Let $n \in \mathbb{N}$ and $\theta \in \mathbb{R}_+$, such that $n \geq 2\theta$. Let $\mathcal{A}_n$ be a consistent learning algorithm. Let $\mathcal{D}$ be a distribution on $\mathcal{Z}$ and assume that, with probability one over a sequence $S = (Z_1, \ldots, Z_{n+1}) \sim \mathcal{D}^{n+1}$, we have $\frac{1}{n+1}\sum_{i=1}^{n+1}\mathbb{E}^S[\ell(\mathcal{A}_n(S_{-i}), Z_i)] \leq \frac{\theta}{n+1}$, where the expectation is taken only over the randomness in $\mathcal{A}_n$. Then,*

$$\mathrm{eCMI}_{\mathcal{D}}(\ell(\mathcal{A}_n)) \leq \theta\log((n+1)/\theta) + 2\theta\log 2.$$

## 5.1 The One-Inclusion Graph Prediction Strategy

Haussler et al. [5] proposed an improper learning rule for learning VC classes based on the one-inclusion graph [34]. We provide a description of this algorithm in Appendix I. The deterministic version of this prediction rule satisfies the following property. Let $\mathcal{H}$ be a concept class with VC dimension $d$. For every $n \in \mathbb{N}$, $h \in \mathcal{H}$, and $(x_1, \ldots, x_{n+1}) \in \mathcal{X}^{n+1}$, let $S = ((x_1, h(x_1)), \ldots, (x_{n+1}, h(x_{n+1})))$. Then $\frac{1}{n+1} \sum_{i=1}^{n+1} \ell(\mathcal{A}_n(S_{-i}), (x_i, h(x_i))) \leq \frac{d}{n+1}$. A direct application of Corollary 5.3 gives the following results.

**Corollary 5.4.** *Let $\mathcal{A}_n$ denote the deterministic one-inclusion graph for learning class $\mathcal{H}$ with VC dimension $d$. Then, for every realizable distribution $\mathcal{D}$ and $n \geq 2d$, we have $\mathrm{eCMI}_{\mathcal{D}}(\ell(\mathcal{A}_n)) \leq d \log((n+1)/d) + 2d \log 2$.*

*Remark* 5.5. In Theorem 2.3 we provide a bound on eCMI of any proper ERM. However, for improper learners, we can construct a consistent algorithm with maximal eCMI. For instance, consider $\mathcal{X} = [0,1]$, $\mathcal{D}_X = \mathrm{Unif}([0,1])$, the concept class of threshold with target function $h^\star(x) = \mathbb{1}[x \geq 1/2]$. Consider a learning algorithm that gives the correct predictions on the points that are in the training set, and for a point that is not in the training set it always predicts one. One can show that eCMI of this consistent algorithm is $\Omega(n)$. ◁

Haussler et al. [5] showed that the one-inclusion graph algorithm achieves $\mathbb{E}R_{\mathcal{D}}(\mathcal{A}_n(S_n)) \leq d/n$ for learning a class $\mathcal{H}$ with VC dimension $d$. Corollary 5.4 implies that $\mathrm{eCMI}_{\mathcal{D}}(\ell(\mathcal{A}_n)) = O(d \log(n))$ for every *deterministic* one-inclusion graph prediction rule. Combining this result with Eq. (3) provides a bound on the excess risk which is suboptimal by a $\log n$ factor. In the next theorem, we show that, in at least one interesting special case, it is possible to remove the logarithmic factor from eCMI by exploiting a *randomized* one-inclusion graph prediction algorithm.

**Theorem 5.6.** *Let $\mathcal{H}$ denote the class of singletons (point functions) on $\mathcal{X} = \mathbb{R}$. There exists a randomized one-inclusion graph prediction rule $\mathcal{A}_n$ for learning class $\mathcal{H}$ such that for every realizable distribution $\mathcal{D}$ and $n \geq 2$, we have $\mathrm{eCMI}_{\mathcal{D}}(\ell(\mathcal{A}_n)) = O(1)$.*

## 6 Remaining Gaps and Open Questions

For proper learning of VC classes, Hanneke [35] showed the assumption $\mathfrak{s} < \infty$ is a necessary and sufficient condition for the existence of a distribution-free bound on the expected risk of all ERMs converging at a rate $O(1/n)$. In Corollary 4.7 and Theorem 4.9 , we showed the same rate for the expected risk of a broad class of ERMs can be obtained using the CMI framework. It is an open question to show that for a class with finite star number, every ERM has bounded eCMI.

An important open problem is to show that for every VC class with finite dual Helly number [2] there exists a proper learning algorithm such that for every data distribution its expected empirical risk converges at a rate of $O(1/n)$ and it has bounded CMI. Combining the generalization guarantees that one can retrieve from Eq. (2) the expected excess risk of the learner with these properties matches the optimal rate from Bousquet et al. [2].

For improper learning of VC classes, we showed a general result for the deterministic one-inclusion graph prediction rule which is suboptimal by a $\log n$ factor. We conjecture that for every VC class with dimension $d$ there exists a probability assignment for the randomized one-inclusion graph for which eCMI is $O(d)$. In Theorem 5.6, we showed this claim holds for the class of point functions.

We also remark that if the answers to the above questions are affirmative, then it can be argued that the CMI framework is expressive enough so that it can explain generalization properties of VC classes. Otherwise, a negative answer to any of the questions implies that there is gap between CMI framework and VC theory.

In Theorem 5.1 we proved that eCMI is *universal* in the realizable setting. A fundamental question to ask is whether for every data distribution $\mathcal{D}$ and consistent learner $\mathcal{A}$, $\mathrm{eCMI}_{\mathcal{D}}(\ell(\mathcal{A}_n))/n$ vanishes as the number training samples $n$ diverges *at the same rate* with the excess risk, i.e., $R_{\mathcal{D}}(\mathcal{A}_n)$.

**Acknowledgments**

The authors would like to thank Blair Bilodeau, Mufan Bill Li, and Jeffery Negrea for feedback on drafts of this work.

**Funding**

M. Haghifam is supported by the Vector Institute, University of Toronto, and a MITACS Accelerate Fellowship with Element AI. S. Moran is a Robert J. Shillman Fellow and is supported by the ISF, grant no. 1225/20, by an Azrieli Faculty Fellowship, and by BSF grant 2018385. D. M. Roy was supported, in part, by an NSERC Discovery Grant, Ontario Early Researcher Award, and a stipend provided by the Charles Simonyi Endowment. Resources used in preparing this research were provided, in part, by the Province of Ontario, the Government of Canada through CIFAR, and companies sponsoring the Vector Institute www.vectorinstitute.ai/partners.

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
