## A  Known Bounds for Learning VC Classes

In this part, we provide a landscape of the known results for the learning VC classes. One key distinction is proper learning versus improper learning. In particular, for every VC class with dimension $d$, there exists a consistent and improper learning algorithm that achieves $O(d/n)$ risk under realizability, and this bound is optimal [36, 5]. The situation for proper learning is much more complicated. In general, the achievable rate for the proper learning of VC classes is off by a log factor, i.e., $O(d\log(n)/n)$. Bousquet et al. [2] show that when the dual Helly and hollow star number, which are combinatorial complexity measures of the class, agree, then they characterize the existence of an optimal proper learner. Also, a subclass of VC for which the log factor provably cannot be removed using proper learners is characterized in [2, Thm. 11]. Moreover, for general ERMs, Hanneke [35] shows the finiteness of star number is a necessary and sufficient condition under which we can remove the log factor using any arbitrary ERMs.

It is interesting to note that the results of Moran and Yehudayoff [37] reveal a connection between the general sample compression schemes and VC classes. However, it is not known whether, in general, the optimal rates for VC classes are always witnessed by compression schemes.

## B  Proof of Theorem 2.3

Fix $n \in \mathbb{N}$. We have $\mathrm{eCMI}_{\mathcal{D}}(\ell(\mathcal{A}_n)) = \mathbb{E}[I^Z(L;U)]$ by definition, and $I^Z(L;U) \leq \mathrm{H}^Z(L)$ a.s., by the nonnegativity of (conditional) entropy. It then suffices to bound the cardinality, $C$, of the support of the conditional distribution of $L$, given $Z$, because $\mathrm{H}^Z(L) \leq \log C$ a.s. For all $i \in \{0,1\}$ and $j \in [n]$, write $Z_{i,j} = (X_{i,j}, Y_{i,j})$, and consider the set of possible predictions on $Z$,

$$P = \{p \in \{0,1\}^{\{0,1\}\times[n]} : \exists h \in \mathcal{H}_n, \forall i \in \{0,1\}, j \in [n], p_{i,j} = h(X_{i,j})\}. \tag{10}$$

The set $P$ is precisely the set of possible labellings of the $2n$ inputs $(X_{i,j})$, which is bounded by the growth function of $\mathcal{H}_n$ evaluated at $2n$ points. By the Sauer–Shelah lemma, the cardinality of $P$ is thus bounded above by $6n^{d_n}$. Note that the support of the conditional distribution of $L$ given $Z$ is

$$\{c \in \{0,1\}^{\{0,1\}\times[n]} : \exists p \in P, \forall i \in \{0,1\}, \forall j \in [n], \text{ if and only if } c_{i,j} = \mathbb{1}[p_{i,j} \neq Y_{i,j}]\}. \tag{11}$$

Therefore, the cardinality of the support is no greater than that of $P$, hence $C \leq 6n^{d_n}$.

## C  Proof of Theorem 4.4

We prove the claim by contradiction. Pick $f$ and $c \geq 0$. Let $\mathcal{H}$ be a concept class with finite VC dimension $d$ as shown to exist by Theorem 4.3.

Let $\mathcal{A}$ be a proper learning algorithm for $\mathcal{H}$, let $n \geq d$, and assume, for the eventual purpose of obtaining a contradiction, that $\mathrm{CMI}_{\mathcal{D}}(\mathcal{A}_n) \leq f(d)$ for all $\mathcal{D}$ and, for all $s \in \mathcal{Z}^n$, $\mathbb{E}\hat{R}_s(\mathcal{A}_n(s)) \leq c\,d/n$ if there exists $h \in \mathcal{H}$ such that $\hat{R}_s(h) = 0$. Pick a realizable distribution $\mathcal{D}$. It follows from the above assumption and Eq. (1) that $\mathbb{E}[R_{\mathcal{D}}(\mathcal{A}_n(S_n))] \leq 2c\,d/n + 3f(d)/n = (3f(d) + 2c\,d)/n$ By Markov's inequality, $\mathbb{P}(R_{\mathcal{D}}(\mathcal{A}_n(S_n)) \geq \epsilon) \leq \frac{1}{n\epsilon}(3f(d) + 2c\,d)$. It follows that the sample complexity of proper learning $\mathcal{H}$ satisfies

$$\mathcal{M}_{\mathrm{prop}}^{\mathcal{H}}(\epsilon, \delta) \leq \frac{1}{\epsilon\delta}(3f(d) + 2c\,d). \tag{12}$$

Now, fix $\delta \in (0, 1/100)$ and fix a convergent sequence of $\epsilon_i \downarrow 0$. There exists $J$ such that, for all $i \geq J$,

$$\frac{1}{\tilde{c}\delta}(3f(d) + 2c\,d) < d\,\mathrm{Log}\frac{1}{\epsilon_i} + \mathrm{Log}\frac{1}{\delta}, \tag{13}$$

for $\tilde{c}$ as in Theorem 4.3. Combining Eq. (13) with Eq. (12), $\mathcal{M}_{\mathrm{prop}}^{\mathcal{H}}(\epsilon_i, \delta) < \frac{\tilde{c}}{\epsilon_i}(d\,\mathrm{Log}\frac{1}{\epsilon_i} + \mathrm{Log}\frac{1}{\delta})$ for $i \geq J$. Simultaneously, from Theorem 4.3, it follows that $\mathcal{M}_{\mathrm{prop}}^{\mathcal{H}}(\epsilon_i, \delta) \geq \frac{\tilde{c}}{\epsilon_i}(d\,\mathrm{Log}\frac{1}{\epsilon_i} + \mathrm{Log}\frac{1}{\delta})$, a contradiction.

# D Proof of Theorem 4.6

Let $n \geq \mathfrak{s}$. First we begin with two definitions: letting $S_n$ be the random element in $(\mathcal{X} \times \mathcal{Y})^n$ representing our training sample, the *empirical teaching dimension* [38, 39], denoted $\mathrm{ETD}_n$, is size of the smallest subset of $S$ that produces the same version space, i.e.,

$$\mathrm{ETD}_n = \min\{|S'| : S' \subseteq S_n, \mathrm{V}_{\mathcal{H}}[S'] = \mathrm{V}_{\mathcal{H}}[S_n]\}. \tag{14}$$

An important fact about the empirical teaching dimension is that $\mathrm{ETD}_n$ is bounded by star number almost surely [35]. An *empirical teaching set* is any subset of $S_n$ that achieves the minimum in the definition of the empirical teaching dimension. Consider a realizable training set $S_n \in (\mathcal{X} \times \mathcal{Y})^n$. Let $S'$ denote an empirical teaching set of $S_n$. Let $\hat{S} \subseteq S_n \setminus S'$. By the definition of the version space we have $\mathrm{V}_{\mathcal{H}}[S_n] \subseteq \mathrm{V}_{\mathcal{H}}[S_n \setminus \hat{S}] \subseteq \mathrm{V}_{\mathcal{H}}[S']$. Also, by the definition of $S'$, we have $\mathrm{V}_{\mathcal{H}}[S_n] = \mathrm{V}_{\mathcal{H}}[S']$; therefore, $\mathrm{V}_{\mathcal{H}}[S_n] = \mathrm{V}_{\mathcal{H}}[S_n \setminus \hat{S}]$. This argument shows that the version space is *stable*, in the sense that removing any point that is not in $S'$ does not alter the version space. Let $\mathrm{ETS}(S_n)$ denote the empirical teaching set $S_n$.

We also need the definition of the *region of disagreement* denoted by $\mathrm{DIS}(\mathrm{V}_{\mathcal{H}}[S_n]) = \{x \in \mathcal{X} | \exists h, h' \in \mathrm{V}_{\mathcal{H}}[S_n]$ such that $h(x) \neq h(x')\}$.

Following the same line of reasoning as in Eq. (4) we obtain $I(\mathrm{V}_{\mathcal{H}}[Z_U]; U|Z) \leq n \log 2 - \sum_{i=1}^{n} \mathrm{H}(U_i|U_{-i}, \mathrm{V}_{\mathcal{H}}[Z_U], Z)$. Since the order of the training set does not change the version space we get $\sum_{i=1}^{n} \mathrm{H}(U_i|U_{-i}, \mathrm{V}_{\mathcal{H}}[Z_U], Z) = n\mathrm{H}(U_1|U_{-1}, \mathrm{V}_{\mathcal{H}}[Z_U], Z)$. Fix $i \in [n]$, and define $U_{i \to b} \triangleq (U_1, \ldots, U_{i-1}, b, U_{i+1}, \ldots, U_n)$ for $b \in \{0, 1\}$. Using this notation we can define training sets $S_{i \to b} = Z_{U_{i \to b}}$ for $b \in \{0, 1\}$ and $i \in [n]$. Let $\mathcal{F}_1 = \sigma(U_{-1}, \mathrm{V}_{\mathcal{H}}[Z_U], Z)$. Then, we have

$$\mathrm{H}(U_1|U_{-1}, \mathrm{V}_{\mathcal{H}}[Z_U], Z) \geq \mathbb{E}\big[\mathrm{H}^{\mathcal{F}_1}(U_1)\mathbb{1}[\mathrm{V}_{\mathcal{H}}[S_{1 \to 0}] = \mathrm{V}_{\mathcal{H}}[S_{1 \to 1}]]\big]. \tag{15}$$

Using the same techniques as in the proof Theorem 3.4, we can show that on the event $\mathbb{1}[\mathrm{V}_{\mathcal{H}}[S_{1 \to 0}] = \mathrm{V}_{\mathcal{H}}[S_{1 \to 1}]], \mathrm{H}^{\mathcal{F}_1}(U_1) = \log(2)$. Thus, $\mathrm{H}(U_1|U_{-1}, \mathrm{V}_{\mathcal{H}}[Z_U], Z) \geq \mathbb{P}(\mathrm{V}_{\mathcal{H}}[S_{1 \to 0}] = \mathrm{V}_{\mathcal{H}}[S_{1 \to 1}]) \log 2$.

For $i \in [n]$, define the training set $S_{-i} \triangleq \{Z_{U_1}, \ldots, Z_{U_{i-1}}, Z_{U_{i+1}}, \ldots, Z_{U_n}\}$. Recall that $Z_{i,j} = (X_{i,j}, Y_{i,j})$. We claim that

$$\mathrm{V}_{\mathcal{H}}[S_{1 \to 0}] \neq \mathrm{V}_{\mathcal{H}}[S_{1 \to 1}] \Rightarrow (X_{0,1} \in \mathrm{DIS}(\mathrm{V}_{\mathcal{H}}[S_{-1}])) \vee (X_{1,1} \in \mathrm{DIS}(\mathrm{V}_{\mathcal{H}}[S_{-1}])). \tag{16}$$

We prove this claim by contraposition. Given that $(X_{0,1} \notin \mathrm{DIS}(\mathrm{V}_{\mathcal{H}}[S_{-1}])) \wedge (X_{1,1} \notin \mathrm{DIS}(\mathrm{V}_{\mathcal{H}}[S_{-1}]))$, we have that the concepts in $\mathrm{DIS}(\mathrm{V}_{\mathcal{H}}[S_{-1}])$ agree for prediction of $X_{0,1}$ and $X_{1,1}$. As $\mathcal{D}$ is a realizable distribution, we conclude $\mathrm{V}_{\mathcal{H}}[S_{1 \to 0}] = \mathrm{V}_{\mathcal{H}}[S_{-1}] = \mathrm{V}_{\mathcal{H}}[S_{1 \to 1}]$.

In the next step, we provide an upper bound on $\mathbb{P}((X_{0,1} \in \mathrm{DIS}(\mathrm{V}_{\mathcal{H}}[S_{-1}])) \vee (X_{1,1} \in \mathrm{DIS}(\mathrm{V}_{\mathcal{H}}[S_{-1}])))$ as follows

$$\begin{aligned}
&\mathbb{P}((X_{0,1} \in \mathrm{DIS}(\mathrm{V}_{\mathcal{H}}[S_{-1}])) \vee (X_{1,1} \in \mathrm{DIS}(\mathrm{V}_{\mathcal{H}}[S_{-1}]))) \\
&\leq \mathbb{P}((X_{0,1} \in \mathrm{DIS}(\mathrm{V}_{\mathcal{H}}[S_{-1}]))) + \mathbb{P}((X_{1,1} \in \mathrm{DIS}(\mathrm{V}_{\mathcal{H}}[S_{-1}]))) \\
&= 2\mathbb{P}(X_{0,1} \in \mathrm{DIS}(\mathrm{V}_{\mathcal{H}}[S_{-1}])).
\end{aligned} \tag{17}$$

Here, we have used the union bound and the points in $Z$ are i.i.d.. Then, we can write

$$\begin{aligned}
\mathbb{P}(X_{0,1} \in \mathrm{DIS}(\mathrm{V}_{\mathcal{H}}[S_{-1}])) &= \mathbb{E}[\mathbb{1}[X_{0,1} \in \mathrm{DIS}(\mathrm{V}_{\mathcal{H}}[S_{-1}])]] \\
&= \mathbb{E}[\mathbb{1}[X_1 \in \mathrm{DIS}(\mathrm{V}_{\mathcal{H}}[\{Z_2, \ldots, Z_n\}])]].
\end{aligned} \tag{18}$$

The last step follows from $U$ and $Z$ are independent, and the points in $Z$ are i.i.d.. Therefore, in the last step we consider the expectation over $(Z_1, \ldots, Z_n) \sim \mathcal{D}^{\otimes n}$. Note that $Z_i = (X_i, Y_i)$. By the exchangeability of the points in $(Z_1, \ldots, Z_n)$ we have

$$\mathbb{E}[\mathbb{1}[X_1 \in \mathrm{DIS}(\mathrm{V}_{\mathcal{H}}[\{Z_2, \ldots, Z_n\}])]] = \frac{1}{n} \sum_{i=1}^{n} \mathbb{E}[\mathbb{1}[X_i \in \mathrm{DIS}(\mathrm{V}_{\mathcal{H}}[\{Z_1, \ldots, Z_n\} \setminus \{Z_i\}])]]. \tag{19}$$

Then, we claim that given $X_i \in \mathrm{DIS}(\mathrm{V}_{\mathcal{H}}[\{Z_1, \ldots, Z_n\} \setminus \{Z_i\}])$ then $X_i \in S'$ where $S'$ is *any* teaching set of $\{Z_1, \ldots, Z_n\}$. We can easily prove this claim by contradiction and the stability of the

version space shown in the beginning of this section. Therefore,

$$\frac{1}{n}\sum_{i=1}^{n}\mathbb{E}[\mathbb{1}[X_i \in \mathrm{DIS}(\mathrm{V}_{\mathcal{H}}[\{Z_1,\dots,Z_n\}\setminus\{Z_i\}])]] \leq \frac{1}{n}\sum_{i=1}^{n}\mathbb{E}[\mathbb{1}[X_i \in S']]$$

$$= \frac{1}{n}\mathbb{E}\left[\sum_{i=1}^{n}\mathbb{1}[X_i \in S']\right]$$

$$\leq \frac{\mathfrak{s}}{n}. \tag{20}$$

Here, we have used the linearity of the expectation, and the last step follows from the fact that the cardinality of $S'$ is at most $\mathfrak{s}$ almost surely. By Eq. (17)-Eq. (20), we have $\mathbb{P}((X_{0,1} \in \mathrm{DIS}(\mathrm{V}_{\mathcal{H}}[S_{-1}])) \vee (X_{1,1} \in \mathrm{DIS}(\mathrm{V}_{\mathcal{H}}[S_{-1}]))) \leq 2\mathfrak{s}/n$. Then, by Eq. (16) we have

$$\mathrm{H}(U_1|U_{-1},\mathrm{V}_{\mathcal{H}}[Z_U],Z) \geq \mathbb{P}(\mathrm{V}_{\mathcal{H}}[S_{1\to 0}] = \mathrm{V}_{\mathcal{H}}[S_{1\to 1}])\log 2$$

$$\geq [1 - \mathbb{P}((X_{0,1} \in \mathrm{DIS}(\mathrm{V}_{\mathcal{H}}[S_{-1}])) \vee (X_{1,1} \in \mathrm{DIS}(\mathrm{V}_{\mathcal{H}}[S_{-1}])))]\log 2$$

$$\geq (1 - \frac{2\mathfrak{s}}{n})\log 2. \tag{21}$$

Finally, combining this results and $I(\mathrm{V}_{\mathcal{H}}[Z_U];U|Z) \leq n\log 2 - n\mathrm{H}(U_1|U_{-1},\mathrm{V}_{\mathcal{H}}[Z_U],Z)$, we obtain $I(\mathrm{V}_{\mathcal{H}}[Z_U];U|Z) \leq 2\mathfrak{s}\log 2$ which was to be shown.

# E  Proof of Theorem 4.8

We begin the proof by a lemma, which can be seen as the generalization of the well-known coupon collector's problem [2, Lem. 19].

**Lemma E.1.** *Let $M,m \in \mathbb{N}$ and $1 \leq m \leq M$. Assume we take $k$ samples $X_1,\dots,X_k$ uniformly at random with replacement from $[M]$. Then $\mathbb{P}(|[M]\setminus\{X_1,\dots,X_k\}| \geq m) \geq 1/2$ if $k \leq (M/2)\log\frac{M}{m}$.*

Note that $|[M]\setminus\{X_1,\dots,X_k\}|$ is the number of unseen elements from $[M]$ after observing samples $X_1,\dots,X_k$.

Consider a concept class $\mathcal{H}$ over the input space $\mathcal{X}$ whose star number is $\mathfrak{s}$. Let $M = \min\{n,\mathfrak{s}\}$. By the definition of star number, there exists $(x_1,\dots,x_M) \in \mathcal{X}^M$, such that $((x_1,y_1),\dots,(x_M,y_M))$ is realizable by $h_0^\star \in \mathcal{H}$, and every neighbour of this sequence is also realizable by a classifier in $\mathcal{H}$. Let $h_{x_i} \in \mathcal{H}$ be any classifier such that $\{j \in [M]|h_{x_i}(x_j) \neq y_j\} = \{i\}$.

For the case $M = 1$, our lower bound is simply $\min\{n,\mathfrak{s}\} - 1 = M - 1 = 0$ which is trivial since the mutual information is non-negative. Therefore, in the rest of the proof we assume $M \geq 2$.

For the case $M \geq 2$, consider the following distribution on the input space

$$\mathcal{D}_X(x_1) = 1 - \frac{M-1}{n} \text{ and } \mathcal{D}_X(x_i) = \frac{1}{n}, \text{ for } i \in \{2,\dots,M\}.$$

Also let the target function (labelling function) be $h_0^\star$. Let $Z,U$, and $S$ be defined as usual, based on a sample from $\mathcal{D}_X$ labeled by $h_0^\star$. Since $\mathcal{D}_X$ has zero measure on $\mathcal{X}\setminus\{x_1,\dots,x_M\}$, we can, without any loss of generality, assume that $\mathcal{H} = \{h_0^\star,h_{x_1},\dots,h_{x_M}\}$, as every other classifier is equivalent to one of these $\mathcal{D}_X$-almost everywhere. When we release the version space, we can agree in advance that elements stand for their equivalence classes, which does not affect the information content.

Let $X = (X_{i,j})_{i\in\{0,1\},j\in[n]}$ denote the inputs observed in the supersample $Z$. Define $X_U$ as the sequence of the inputs observed in $Z_U$. Similarly, let $X_{\bar{U}}$ denote the sequence of inputs observed in the "ghost sample" $Z_{\bar{U}}$, where $\bar{U}$ denotes the sequence $U$ but with every entry flipped. In the following, we write $x \in X_U$ to mean that there exists $i \in [n]$ such that $X_{U_i,i} = x$.

We claim that $\mathrm{V}_{\mathcal{H}}[Z_U] = \{h_{x_i}|i \in [M], x_i \notin X_U\} \cup \{h_0^\star\}$.[5] To see this, note that, if $x_i \notin X_U$, then $h_{x_i} = h_0^\star$ on $S$ and so both are ERMs. In the other direction, if $x_i \in X_U$, then $h_{x_i}$ makes at least one mistake, and so is not an ERM. Thus, if $\mathrm{V}_{\mathcal{H}}[Z_U]$ has a non-empty intersection with $\{h_{X_{0,i}},h_{X_{1,i}}\}$,

---

[5]Formally, this statement holds almost surely. We will skip such declarations for the remainder of the proof.

we can perfectly recover $U_i$ from $V_{\mathcal{H}}[Z_U]$, since in each column of the supersample $Z$, only one point is selected for the training set. We use this observation to lower bound the conditional entropy of $U$ given $Z$ and the version space.

To that end, let $J = \{j \in [n] | V_{\mathcal{H}}[Z_U] \cap \{h_{X_{0,j}}, h_{X_{1,j}}\} \neq \emptyset\}$ be the set that contains the index of every column $j$ for which we can perfectly recover $U_j$ from the version space. Note that we can represent $J$ in an equivalent form as follows. For two sequences $v, w \in \{x_1, \ldots, x_M\}^n$, define $K(v, w) = \{j \in [n] | w_j \neq v_i \text{ for all } i \in [n]\}$. By the definition of the version space, $x_i \notin X_U$ is equivalent to $h_{x_i} \in V_{\mathcal{H}}[Z_U]$. Let $j \in [n]$, then

$$j \in J \Leftrightarrow V_{\mathcal{H}}[Z_U] \cap \{h_{X_{1-U_j,j}}\} = \{h_{X_{1-U_j,j}}\} \Leftrightarrow X_{1-U_j,j} \notin X_U \Leftrightarrow j \in K(X_U, X_{\bar{U}}).$$

Therefore, $J = K(X_U, X_{\bar{U}})$.

By the definition of the mutual information, and the fact that $U$ and $Z$ are independent, we have

$$I(V_{\mathcal{H}}[Z_U]; U|Z) = n \log(2) - H(U|Z, V_{\mathcal{H}}[Z_U]). \tag{22}$$

Then

$$H(U|Z, V_{\mathcal{H}}[Z_U]) = H(U|Z, V_{\mathcal{H}}[Z_U], J) \tag{23}$$
$$= H(U_J, U_{J^c}|Z, V_{\mathcal{H}}[Z_U], J) \tag{24}$$
$$= H(U_{J^c}|Z, V_{\mathcal{H}}[Z_U], J) \tag{25}$$
$$\leq \mathbb{E}[n - |J|] \log(2), \tag{26}$$

where Eq. (23) follows from $J$ being known from $Z$ and $V_{\mathcal{H}}[Z_U]$; Eq. (25) follows from $U_J$ being known from $J, Z, V_{\mathcal{H}}[Z_U]$; and Eq. (26) follows from the cardinality of the support of the distribution of $U_{J^c}$ being no more than $2^{(n-|J|)}$. Therefore, by Eq. (22) and Eq. (26), we have

$$I(V_{\mathcal{H}}[Z_U]; U|Z) \geq \mathbb{E}[J] \log 2. \tag{27}$$

In the next step of the proof, we lower bound $\mathbb{E}[|J|]$. Because $U$ and $Z$ are independent and $Z$ is an i.i.d. array, $X_U$ and $X_{\bar{U}}$ are independent and identically distributed sequences of i.i.d. elements with common distribution $\mathcal{D}_{\mathcal{X}}$. Thus, $\mathbb{E}[|J|] = \mathbb{E}[|K(X, \bar{X})|]$, where $X$ and $\bar{X}$ are i.i.d. copies of $X_U$ (equivalently, $X_{\bar{U}}$).

Let $\hat{M}$ be the number of elements of $\{x_2, \ldots, x_M\}$ *not* appearing in $X$, i.e., $\hat{M} = |\{i \in \{2, \ldots, M\} | x_i \neq X_j \text{ for all } j \in [n]\}|$. Conditional on $X$, if $x_1 \in X$, then $|K(X, \bar{X})|$ is a Binomial random variable with $n$ trials, each succeeding with probability $\frac{\hat{M}}{n}$. If $x_1 \notin X$, then $|K(X, \bar{X})|$ is a Binomial random variable with $n$ trials, each succeeding with probability $\frac{\hat{M}}{n} + 1 - \frac{M-1}{n} \geq \frac{\hat{M}}{n}$. Therefore,

$$\mathbb{E}[|J|] = \mathbb{E}[|K(X, \bar{X})|] \tag{28}$$
$$= \mathbb{E}[\mathbb{E}^X[|K(X, \bar{X})|]] \tag{29}$$
$$\geq \mathbb{E}[\hat{M}]. \tag{30}$$

Here, Eq. (29) follows from the tower rule and Eq. (30) follows from the fact that the mean of the binomial distribution with $n$ trials, each succeeding with probability $p$, is $np$.

Fix $\beta = \exp(-3)$ and $m = \beta(M - 1)$. Let $\hat{N}$ denote the number of samples in $X$ falling in $\{x_2, \ldots, x_M\}$. Then, using the tower rule and Markov's inequality we have

$$\mathbb{E}[\hat{M}] = \mathbb{E}[\mathbb{E}^{\hat{N}}[\hat{M}]] \tag{31}$$
$$\geq \mathbb{E}[m\mathbb{P}^{\hat{N}}[\hat{M} \geq m]] \tag{32}$$
$$\geq \mathbb{E}[m\mathbb{P}^{\hat{N}}[\hat{M} \geq m]\mathbb{1}[\hat{N} \leq ((M - 1)/2)\log \beta^{-1}]] \tag{33}$$

Note that, conditional on $\hat{N}$, the $\hat{N}$ samples falling in $\{x_2, \ldots, x_M\}$ are conditionally independent, with conditional distribution uniform on this set. Using this fact, from Lemma E.1 we obtain

$$\mathbb{E}[m\mathbb{P}^{\hat{N}}[\hat{M} \geq m]\mathbb{1}[\hat{N} \leq ((M - 1)/2)\log \beta^{-1}]] \geq \frac{1}{2}m\mathbb{P}(\hat{N} \leq ((M - 1)/2)\log \beta^{-1}). \tag{34}$$

We have $\mathbb{E}[\hat{N}] = M - 1$ and, by Markov's inequality, $\mathbb{P}(\hat{N} \leq \frac{M-1}{2} \log \beta^{-1}) \geq 1 - 2/\log(\beta^{-1}) = 1/3$. By combining Eq. (27), Eq. (30), Eq. (34), and $\mathbb{P}(\hat{N} \leq \frac{M-1}{2} \log \beta^{-1}) \geq 1/3$ we get

$$I(\mathrm{V}_{\mathcal{H}}[Z_U]; U | Z) \geq \frac{\beta \log 2}{6}(M - 1)$$
$$= \Omega(\min\{n, \mathfrak{s}\} - 1),$$

which was to be shown.

# F   Proof of Theorem 4.9

By the well-ordering theorem [40], there exists a binary relation $\ll$ on $\mathcal{H}$ that is transitive, total, antisymmetric, and well-ordered. In particular, the well-ordered property implies that every nonempty subset $\mathcal{H}$ has the least element. The proposed learning algorithm is given by

$$\mathcal{A}_n(S_n) = \mathrm{LE}(\mathrm{V}_{\mathcal{H}}[S_n]),$$

where LE of a nonempty set denotes its least element with respect to $\ll$.[6] Note that $\mathcal{A}_n$ is *deterministic*, *consistent*, and *permutation-invariant*, i.e., the order of the points in $S_n$ does not impact the output. Let $W = \mathcal{A}_n(Z_U)$. By the definition of the mutual information and Lemma 2.1, we obtain

$$\mathrm{CMI}_{\mathcal{D}}(\mathcal{A}_n) = H(U | Z) - H(U | W, Z)$$

$$\leq n \log(2) - \sum_{i=1}^{n} H(U_i | U_{-i}, Z, W). \tag{35}$$

The last step follows since $U$ is independent of $Z$ and the independence of indices of $U$ and Lemma 2.1. By the permutation-invariance of the algorithm, we have $\sum_{i=1}^{n} H(U_i | U_{-i}, Z, W) = nH(U_1 | U_{-1}, Z, W)$. Fix $i \in [n]$, and define $U_{i \to j} \triangleq (U_1, \dots, U_{i-1}, j, U_{i+1}, \dots, U_n)$ for $j \in \{0, 1\}$. Using this notation we can define training set $S_{i \to j} = Z_{U_{i \to j}}$ for $j \in \{0, 1\}$ and $U_{-i} \in \{0, 1\}^{n-1}$. Let $\mathcal{F}_1 = \sigma(W, U_{-1}, Z)$. Define the $\mathcal{F}_1$-measurable event $\mathcal{E} = \{\mathcal{A}_n(S_{1 \to 0}) = \mathcal{A}_n(S_{1 \to 1})\}$. Then, we can write $H(U_1 | U_{-1}, Z, W) = \mathbb{E}[\mathrm{H}^{\mathcal{F}_1}(U_1)(\mathbb{1}[\mathcal{E}] + \mathbb{1}[\mathcal{E}^c])]$. We claim the following facts:

1. On the event $\mathcal{E}^c$, $\mathrm{H}^{\mathcal{F}_1}(U_1) = 0$ a.s..
2. On the event $\mathcal{E}$, $\mathrm{H}^{\mathcal{F}_1}(U_1) = \log 2$ a.s..

The reason is that since the algorithm is deterministic, on the event $\mathcal{E}^c$ we can perfectly recover $U_1$ since $W$ is either $\mathcal{A}_n(S_{1 \to 0})$ or $\mathcal{A}_n(S_{1 \to 1})$ which shows that $\mathrm{H}^{\mathcal{F}_1}(U_1) = 0$. Then, on the event $\mathcal{E}$, using the Bayes rule we can show that $\mathrm{H}^{\mathcal{F}_1}(U_1) = \log 2$. Therefore, we have $H(U_1 | U_{-1}, Z, W) = \mathbb{E}[\mathbb{1}[\mathcal{E}]] \log(2)$. We can further lower bound

$$H(U_1 | U_{-1}, Z, W) \geq \mathbb{E}\left[\mathbb{1}[\mathcal{E}] \mathbb{1}[\ell(\mathcal{A}_n(S_{1 \to 0}), Z_{1,1}) = 0 \wedge \ell(\mathcal{A}_n(S_{1 \to 1}), Z_{0,1}) = 0]\right] \log(2). \tag{36}$$

Next, we claim that on the event $\{\ell(\mathcal{A}_n(S_{1 \to 0}), Z_{1,1}) = 0\} \wedge \{\ell(\mathcal{A}_n(S_{1 \to 1}), Z_{0,1}) = 0\}$, we have $\mathcal{A}_n(S_{1 \to 0}) = \mathcal{A}_n(S_{1 \to 1})$ a.s.. This claim can be proved by contradiction. Assume $\mathcal{A}_n(S_{1 \to 0}) \neq \mathcal{A}_n(S_{1 \to 1})$. Consider the version space $\mathrm{V}_{\mathcal{H}}[S_{1 \to 0} \cup S_{1 \to 1}]$. By the assumptions $\ell(\mathcal{A}_n(S_{1 \to 0}), Z_{1,1}) = 0$ and $\ell(\mathcal{A}_n(S_{1 \to 1}), Z_{0,1}) = 0$, we have $\mathcal{A}_n(S_{1 \to 0}) \in \mathrm{V}_{\mathcal{H}}[S_{1 \to 0} \cup S_{1 \to 1}]$ and $\mathcal{A}_n(S_{1 \to 1}) \in \mathrm{V}_{\mathcal{H}}[S_{1 \to 0} \cup S_{1 \to 1}]$. Also, it is immediate to see that $\mathrm{V}_{\mathcal{H}}[S_{1 \to 0} \cup S_{1 \to 1}] \subseteq \mathrm{V}_{\mathcal{H}}[S_{1 \to 0}]$ and $\mathrm{V}_{\mathcal{H}}[S_{1 \to 0} \cup S_{1 \to 1}] \subseteq \mathrm{V}_{\mathcal{H}}[S_{1 \to 1}]$. Therefore, we have $\mathcal{A}_n(S_{1 \to 0}) \in \mathrm{V}_{\mathcal{H}}[S_{1 \to 1}]$ and $\mathcal{A}_n(S_{1 \to 1}) \in \mathrm{V}_{\mathcal{H}}[S_{1 \to 0}]$. By the definition of the algorithm, we choose the least hypothesis from the version space. Thus, $\mathcal{A}_n(S_{1 \to 0}) \ll \mathcal{A}_n(S_{1 \to 1})$ since $\mathcal{A}_n(S_{1 \to 1}) \in \mathrm{V}_{\mathcal{H}}[S_{1 \to 0}]$. Similarly, we can show $\mathcal{A}_n(S_{1 \to 1}) \ll \mathcal{A}_n(S_{1 \to 0})$. Therefore, considering $\mathcal{A}_n(S_{1 \to 1}) \ll \mathcal{A}_n(S_{1 \to 0})$ and $\mathcal{A}_n(S_{1 \to 0}) \ll \mathcal{A}_n(S_{1 \to 1})$, we conclude that the assumption $\mathcal{A}_n(S_{1 \to 0}) \neq \mathcal{A}_n(S_{1 \to 1})$ is false, a contradiction.

Having proved that $\mathcal{A}_n(S_{1 \to 0}) = \mathcal{A}_n(S_{1 \to 1})$ on the event $\{\ell(\mathcal{A}_n(S_{1 \to 0}), Z_{1,1}) = 0\} \wedge \{\ell(\mathcal{A}_n(S_{1 \to 1}), Z_{0,1}) = 0\}$, we can further simplify Eq. (36) as

$$H(U_1 | U_{-1}, Z, W) \geq \mathbb{E}\left[\mathbb{1}[\ell(\mathcal{A}_n(S_{1 \to 1}), Z_{0,1}) = 0] \mathbb{1}[\ell(\mathcal{A}_n(S_{1 \to 0}), Z_{1,1}) = 0]\right] \log 2$$
$$\geq (1 - \mathbb{E}[\mathbb{1}[\ell(\mathcal{A}_n(S_{1 \to 1}), Z_{0,1}) = 1]] - \mathbb{E}[\mathbb{1}[\ell(\mathcal{A}_n(S_{1 \to 1}), Z_{0,1}) = 1]]) \log 2$$
$$= (1 - 2\mathbb{E}[R_{\mathcal{D}}(\mathcal{A}_n(S_n))]) \log 2.$$

---

[6]In some cases this classifier may not be measurable. We will assume it is. To avoid measure-theoretic issues, one may assume $\mathcal{X}$ is countably infinite or finite, or design a well-ordering by hand to guarantee measurability if possible.

The last step follows since the elements of $S_{1\to 0}$ and $S_{1\to 1}$ are i.i.d.. By plugging this lower bound into Eq. (35), we obtain

$$\mathrm{CMI}_{\mathcal{D}}(\mathcal{A}_n) \leq 2n\log(2)\,\mathbb{E}[R_{\mathcal{D}}(\mathcal{A}_n(S_n))]. \tag{37}$$

Then, we use the the result from Corollary 12 of [35] which states the following bound holds uniformly for the expected risk of *all* consistent and proper learners for learning a concept class with VC dimension $d$ and star number $\mathfrak{s}$:

$$\mathbb{E}[R_{\mathcal{D}}(\mathcal{A}_n(S_n))] = O(\frac{d}{n}\log(\frac{\min\{\mathfrak{s},n\}}{d})).$$

Finally, the stated result follows by combining this bound with Eq. (37).

## G  Proof of Theorem 5.1

Let $L = (L_1,\ldots,L_n)$ where $L_i \in \{0,1\}^2$ denotes the vector at Column $i$ of the loss array $L$. By the definition of mutual information, we have $I(L;U|Z) \leq \mathrm{H}(L|Z)$. Since conditioning decreases the entropy, we have $\mathrm{H}(L|Z) \leq \mathrm{H}(L)$. Finally, by the chain rule, $\mathrm{H}(L) \leq \sum_{i=1}^n \mathrm{H}(L_i)$.

Note that $L_i$ takes values in $\{[1,0]^\intercal, [0,0]^\intercal, [0,1]^\intercal\}$, as it is assumed that $\mathcal{A}_n$ is consistent and, by construction, at each column, one of the points is selected for the training set. If $U_i = 0$, then $Z_{0,i}$ is in the training set and so, because the algorithm is consistent, $L_{0,i} = 0$ a.s. Therefore,

$$\begin{aligned}
\mathbb{P}[L_i = [1,0]^\intercal] &= \mathbb{E}\big[\mathbb{P}^U[L_i = [1,0]^\intercal]\big] \\
&= \mathbb{E}\big[\mathbb{1}[U_i = 0]\mathbb{P}^U[L_i = [1,0]^\intercal]\big] + \mathbb{E}\big[\mathbb{1}[U_i = 1]\mathbb{P}^U[L_i = [1,0]^\intercal]\big] \\
&= \mathbb{E}\big[\mathbb{1}[U_i = 1]\mathbb{P}^U[L_i = [1,0]^\intercal]\big]
\end{aligned}$$

Conditioning on event $U_i = 1$, we have $L_i = [1,0]^\intercal = \ell(\mathcal{A}_n(Z_U), Z_{0,i}) = 1$ and therefore

$$\mathbb{P}^U[L_i = [1,0]^\intercal] = \mathbb{P}^U[\ell(\mathcal{A}_n(Z_U), Z_{0,i}) = 1] = R_{\mathcal{D}}(\mathcal{A}_n),$$

where the final equality follows from the definition of the expected risk and the fact that $Z_{0,i}$ is not in $Z_U$. Therefore, $\mathbb{P}[L_i = [1,0]^\intercal] = \mathbb{E}[\mathbb{1}[U_i = 1]]R_{\mathcal{D}}(\mathcal{A}_n) = (1/2)R_{\mathcal{D}}(\mathcal{A}_n)$. The same idea establishes that $\mathbb{P}[L_i = [0,1]^\intercal] = (1/2)R_{\mathcal{D}}(\mathcal{A}_n)$.

Therefore, by the definition of the entropy

$$\begin{aligned}
\mathrm{H}(L_i) &= -(1 - R_{\mathcal{D}}(\mathcal{A}_n))\log(1 - R_{\mathcal{D}}(\mathcal{A}_n)) - R_{\mathcal{D}}(\mathcal{A}_n)\log(\frac{R_{\mathcal{D}}(\mathcal{A}_n)}{2}) \\
&= \mathrm{H}_b(R_{\mathcal{D}}(\mathcal{A}_n)) + R_{\mathcal{D}}(\mathcal{A}_n)\log(2).
\end{aligned}$$

The stated results follows from $\mathrm{eCMI}_{\mathcal{D}}(\ell(\mathcal{A}_n)) \leq \sum_{i=1}^n \mathrm{H}(L_i)$.

## H  Proof of Theorem 5.6

We begin by presenting a theorem that will be used later to prove Theorem 5.6. This theorem, which might be of independent interest, shows the *average leave-one-error over supersample $Z$* can be used to upper bound $\mathrm{eCMI}_{\mathcal{D}}(\ell(\mathcal{A}_n))$.

Lets begin by introducing some notations. For $n \in \mathbb{N}$, let $[2n]_{n+1}$ denote the set of all size-$n+1$ subsets of $[2n]$ and let $\mathrm{H}_b(\cdot)$ denote the binary entropy function.

**Theorem H.1.** *Let $\mathcal{A}$ denote a consistent and symmetric algorithm. Let $P_e : \mathcal{Z}^n \times \mathcal{Z} \to [0,1]$. For $s = ((x_1,y_1),\ldots,(x_n,y_n)) \in \mathcal{Z}^n$ and $(x,y) \in \mathcal{Z}$, $P_e(s;(x,y))$ denotes the probability of error $\mathcal{A}_n$ for predicting the label $x$ where the randomness is over the internal randomness in $\mathcal{A}$. Then, for every distribution $\mathcal{D}$, we have*

$$\mathrm{eCMI}_{\mathcal{D}}(\ell(\mathcal{A}_n)) \leq n\mathbb{E}\Big[\mathrm{H}_b(\kappa(\tilde{Z})) + \kappa(\tilde{Z})\log 2 - \mathbb{E}_{J \sim \mathit{Unif}([2n]_{n+1})}\frac{1}{n+1}\sum_{j \in J}\mathrm{H}_b(P_e(\tilde{Z}_{J-\{j\}};\tilde{Z}_j))\Big] \tag{38}$$

*where $\tilde{Z} = (Z_1,\ldots,Z_{2n}) \sim \mathcal{D}^{\otimes(2n)}$ and $\kappa(\tilde{Z}) = \mathbb{E}_{J \sim \mathit{Unif}([2n]_{n+1})}\frac{1}{n+1}\sum_{j \in J}P_e(\tilde{Z}_{J-\{j\}};\tilde{Z}_j)$ which takes values in $[0,1]$ almost surely. Note, in Eq. (38), the outer expectation is over $\tilde{Z}$.*

The proof of Theorem H.1 is deferred to Appendix H.2.

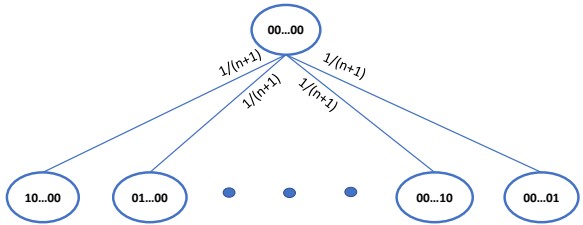

Figure 1: One-inclusion graph of point functions for a set of distinct points.

## H.1  Proof of Theorem 5.6

First of all, note that given *distinct* points $(x_1, \ldots, x_{n+1}) \in \mathbb{R}^{n+1}$, the vertex set of the one-inclusion graph is given by

$$\{(h(x_1), \ldots, h(x_{n+1}))|h \in \mathcal{H}\} = \{(a_1, \ldots, a_{n+1})|a_i \in \{0,1\} \text{ for all } i \in [n+1], \sum_{i \in [n+1]} a_i \leq 1\}.$$

As an example Fig. 1 illustrates the structure of the one-inclusion graph for a sequence of *distinct* points. The prediction rule is as follows. Given a training set $((x_1, y_1), \ldots, (x_n, y_n))$ and test example $x$, we have two cases. If $y_i = 0$ for all $i \in [n]$, then the label of $x$ is predicted to be $0$ with probability $\frac{n}{n+1}$, and it is predicted to be one with probability $\frac{1}{n+1}$. The second case is that there exists a point with label 1. Denote its index by $i^\star \in [n]$. In this case, the target function is known to be $\mathbb{1}[x = x_{i^*}]$.

To upper bound eCMI for this prediction rule, we consider the upper bound provided in Theorem H.1. For a fixed supersample $\tilde{Z} = ((x_1, y_1), \ldots, (x_{2n}, y_{2n}))$, we will consider two cases:

Case I: For all $i \in [2n]$ we have $y_i = 0$. Let $J \in [2n]_{n+1}$ and let $\deg(\tilde{Z}_J)$ denote the degree of the vertex $(0, 0, \ldots, 0, 0)$ in the one-inclusion graph $\tilde{Z}_J$. Note that if there is no repetition in $\tilde{Z}_J$ then, the degree is $n+1$. If repetition exists, it is less than $n+1$. Then, according to the prediction rule we have

$$\kappa(\tilde{Z}) = \mathbb{E}_{J \sim \text{Unif}([2n]_{n+1})} \frac{1}{n+1} \sum_{j \in J} P_e(\tilde{Z}_{J-\{j\}}, \tilde{Z}_j)$$

$$= \frac{1}{n+1} \mathbb{E}_{J \sim \text{Unif}([2n]_{n+1})} \frac{\deg(\tilde{Z}_J)}{n+1}. \tag{39}$$

Also,

$$\mathbb{E}_{J \sim \text{Unif}([2n]_{n+1})} \frac{1}{n+1} \sum_{j \in J} \mathrm{H}_b(P_e(\tilde{Z}_{J-\{j\}}, \tilde{Z}_j)) = \mathrm{H}_b(\frac{1}{n+1}) \mathbb{E}_{J \sim \text{Unif}([2n]_{n+1})} \frac{\deg(\tilde{Z}_J)}{n+1}. \tag{40}$$

let $\alpha = \mathbb{E}_{J \sim \text{Unif}([2n]_{n+1})} \frac{\deg(\tilde{Z}_J)}{n+1}$. If $\alpha = 0$, then $\text{eCMI}_{\mathcal{D}}(\ell(\mathcal{A}_n)) = 0 = O(1)$. Therefore, assume $\alpha > 0$, then we can use Eq. (38), Eq. (39), and Eq. (40) to obtain

$$\text{eCMI}_{\mathcal{D}}(\ell(\mathcal{A}_n)) \leq n\mathbb{E}\Big[\mathrm{H}_b(\frac{\alpha}{n+1}) + \frac{\alpha \log 2}{n+1} - \alpha \mathrm{H}_b(\frac{1}{n+1})\Big] \tag{41}$$

$$\leq n\mathbb{E}\Big[-\frac{\alpha}{n+1}\log\frac{\alpha}{n+1} - (1 - \frac{\alpha}{n+1})\log(1 - \frac{\alpha}{n+1}) + \frac{\alpha \log 2}{n+1} +$$

$$\frac{\alpha}{n+1}\log(\frac{1}{n+1}) + \alpha(1 - \frac{1}{n+1})\log(1 - \frac{1}{n+1})\Big] \tag{42}$$

$$\leq n\mathbb{E}\Big[-\frac{\alpha}{n+1}\log(\alpha) + \frac{2\alpha \log 2}{n+1}\Big] \tag{43}$$

$$\leq n\Big[\frac{2\log 2}{n+1} + \frac{\exp(-1)}{n+1}\Big] \tag{44}$$

$$= O(1). \tag{45}$$

Eq. (43) is obtained by removing the negative terms and the inequality $-(1-x)\log(1-x) \leq x \log 2$ for $x \in [0,1]$. Eq. (44) follows from $\max_{x \in [0,1]} -x\log(x) = \exp(-1)$ and $\alpha \leq 1$.

Case II: There exists $i \in [2n]$ such that $y_i = 1$. Let $m = |\{i \in [2n] | y_i = 1\}|$, $\beta = \frac{\binom{2n-m}{n+1}}{\binom{2n}{n+1}}$, and $\alpha = \mathbb{E}_J[\frac{\deg(Z_J)}{n+1} | \text{no point with label 1 in } \tilde{Z}_J]$. Then, we have

$$\kappa(\tilde{Z}) = \mathbb{E}_{J \sim \text{Unif}([2n]_{n+1})} \frac{1}{n+1} \sum_{j \in J} P_e(\tilde{Z}_{J-\{j\}}, \tilde{Z}_j)$$

$$= \beta \mathbb{E}_J[\frac{1}{n+1} \sum_{j \in J} P_e(\tilde{Z}_{J-\{j\}}, \tilde{Z}_j) | \text{no point with label 1 in } \tilde{Z}_J]$$

$$+ (1-\beta) \mathbb{E}_J[\frac{1}{n+1} \sum_{j \in J} P_e(\tilde{Z}_{J-\{j\}}, \tilde{Z}_j) | \text{there exists points with label 1 in } \tilde{Z}_J]$$

$$\leq \alpha\beta \frac{1}{n+1} + (1-\beta) \frac{1}{(n+1)^2}, \tag{46}$$

where in the last step we have used the fact that when in $\tilde{Z}_J$ there is a point with label 1, then the leave-one-out error is at most $\frac{1}{(n+1)^2}$ and $\beta = \mathbb{P}(\text{no point with label 1 in } \tilde{Z}_J) = \frac{\binom{2n-m}{n+1}}{\binom{2n}{n+1}}$.

The binary entropy function is concave and increasing in the interval $[0, 1/2]$. The concavity of the binary entropy function implies that $H_b(v_2) \leq H_b(v_1)'(v_2 - v_1) + H_b(v_1)$ for all $v_1$ and $v_2$ in $(0, 1)$. Then,

$$H_b(\kappa(\tilde{Z})) \leq H_b(\alpha\beta \frac{1}{n+1} + (1-\beta) \frac{1}{(n+1)^2})$$

$$\leq H_b(\frac{\alpha\beta}{n+1}) + \frac{1-\beta}{(n+1)^2} \log(\frac{n+1-\alpha\beta}{\alpha\beta}). \tag{47}$$

Then, by considering the summation only over $\tilde{Z}_J$ without any point with label 1, we obtain

$$\mathbb{E}_{J \sim \text{Unif}([2n]_{n+1})} \frac{1}{n+1} \sum_{j \in J} H_b(P_e(\tilde{Z}_{J-\{j\}}, \tilde{Z}_j)) \geq \alpha\beta H_b(\frac{1}{n+1}). \tag{48}$$

Then, we can substitute Eq. (47) and Eq. (48) into Eq. (38) to obtain $\text{eCMI}_{\mathcal{D}}(\ell(\mathcal{A}_n)) = O(1)$ following the same line of reasoning as in Eq. (41)–Eq. (45).

## H.2  Proof of Theorem H.1

We begin with introducing some notations. Let $\Gamma_{2n}$ be the set of all bijective mappings from $[2n]$ to $\{0, 1\} \times [n]$. Let $\sigma \in \Gamma_{2n}$ and $x = (x_1, \ldots, x_{2n}) \in \mathcal{X}^{2n}$ be a vector of length $2n$. Then, $x^\sigma$ denotes a matrix of size $2 \times n$ where $x_{i,j}^\sigma = x_{\sigma^{-1}(i,j)}$ for $i \in \{0, 1\}$ and $j \in [n]$. Also, for every $m, k \in \mathbb{N}$ and $1 \leq k \leq m$, let $[m]_k$ denote the set of all subsets of size $k$ of $[m]$.

For every $n \in \mathbb{N}$ let $\pi \sim \text{Unif}(\Gamma_{2n})$, $\tilde{Z} = (Z_1, \ldots, Z_{2n}) \sim \mathcal{D}^{\otimes(2n)}$, and $U = (U_1, \ldots, U_n) \sim (\text{Ber}(\{0, 1\})^{\otimes n}$ where $\pi$, $\tilde{Z}$ and $U$ are mutually independent. Let $S_n = (\tilde{Z}_{U_j,j}^\pi)_{j=1}^n$ and $\mathcal{A}_n(S_n)$ be a learning algorithm. Let $L \in \{0, 1\}^{2 \times n}$ be a matrix with entries $L_{i,j} = \ell(\mathcal{A}_n(S_n), \tilde{Z}_{i,j}^\pi)$ for $i \in \{0, 1\}$ and $j \in [n]$. Then, using these random variables, we can define $I(L; U | \tilde{Z}, \pi)$.

**Lemma H.2.** $\text{eCMI}_{\mathcal{D}}(\ell(\mathcal{A}_n)) = I(L; U | \tilde{Z}, \pi)$

*Proof.* $\tilde{Z}^\pi$ is a $\sigma(\tilde{Z}, \pi)$-measurable random variable therefore we have $I(L; U | \tilde{Z}, \pi) = I(L; U | \tilde{Z}^\pi, \tilde{Z}, \pi)$. Then, conditioned on $\tilde{Z}^\pi$, $L$ and $U$ are independent from $\tilde{Z}$ and $\pi$. Finally, note that the samples in $\tilde{Z}$ are i.i.d., therefore we have $I(L; U | \tilde{Z}^\pi) = \text{eCMI}_{\mathcal{D}}(\ell(\mathcal{A}_n))$. $\square$

**Lemma H.3.** *Let $\pi$, $U$, and $\tilde{Z}$ be as defined in the beginning of this section. Let $n \in \mathbb{N}$ and $f : \mathcal{Z}^n \times \mathcal{Z} \to \mathbb{R}$ be a real-valued function where $f$ is permutation-invariant with respect to its first input. Then*

$$\mathbb{E}^{\tilde{Z}}\left[f((\tilde{Z}_{U_1,1}^\pi, \ldots, \tilde{Z}_{U_n,n}^\pi); \tilde{Z}_{\bar{U}_1,1}^\pi)\right] = \mathbb{E}_{J \sim \text{Unif}([2n]_{n+1})} \frac{1}{n+1} \sum_{j \in J} f(\tilde{Z}_{J-\{j\}}; \tilde{Z}_j), \tag{49}$$

*where $\tilde{Z}_J = (\tilde{Z}_{J_1}, \ldots, \tilde{Z}_{J_{n+1}})$ and $\bar{U}_i = 1 - U_i$.*

*Proof.* write

$$\mathbb{E}^{\tilde{Z}}\left[f(\{\tilde{Z}^{\pi}_{U_1,1},\ldots,\tilde{Z}^{\pi}_{U_n,n}\};\tilde{Z}^{\pi}_{\bar{U}_1,1})\right] =$$

$$\sum_{(i_1,\ldots,i_{n+1})\in[2n]_{n+1}} f((\tilde{Z}_{i_1},\ldots,\tilde{Z}_{i_n});\tilde{Z}_{i_{n+1}})\mathbb{P}^{\tilde{Z}}[(\{\tilde{Z}^{\pi}_{U_1,1},\ldots,\tilde{Z}^{\pi}_{U_n,n}\},\tilde{Z}^{\pi}_{\bar{U}_1,1}) = (\{\tilde{Z}_{i_1},\ldots,\tilde{Z}_{i_n}\},\tilde{Z}_{i_{n+1}})].$$

Then,

$$\mathbb{P}^{\tilde{Z}}[(\{\tilde{Z}^{\pi}_{U_1,1},\ldots,\tilde{Z}^{\pi}_{U_n,n}\},\tilde{Z}^{\pi}_{\bar{U}_1,1}) = (\{\tilde{Z}_{i_1},\ldots,\tilde{Z}_{i_n}\},\tilde{Z}_{i_{n+1}})] = \frac{n!(n-1)!}{(2n)!}$$

$$= \frac{1}{\binom{2n}{n+1}(n+1)}$$

where the last line follows since $U$ and $\pi$ are independent. It is easy to verify that is exactly equal to the RHS of Eq. (49). $\qquad\square$

Let $L = (L_1,\ldots,L_n)$ where $L_i \in \{0,1\}^2$ denotes the vector at Column $i$ of the loss vector $L$. By the definition and the chain rule for mutual information we have

$$I(L;U|\tilde{Z},\pi) = \sum_{i=1}^{n} I(L_i;U|L_1,\ldots,L_{i-1},\tilde{Z},\pi) \tag{50}$$

$$= \sum_{i=1}^{n} \mathrm{H}(L_i|L_1,\ldots,L_{i-1},\tilde{Z},\pi) - \mathrm{H}(L_i|L_1,\ldots,L_{i-1},\tilde{Z},\pi,U) \tag{51}$$

$$\leq \sum_{i=1}^{n} \mathrm{H}(L_i|\tilde{Z},\pi) - \mathrm{H}(L_i|\tilde{Z},\pi,U) \tag{52}$$

$$= n(\mathrm{H}(L_1|\tilde{Z},\pi) - \mathrm{H}(L_1|\tilde{Z},\pi,U)). \tag{53}$$

Here, Eq. (52) due to the Markov chain $L_i - (U,\tilde{Z},\pi) - L_{-i}$ and removing conditions increases the entropy. Then, the last step follows since the algorithm is permutation-invariant. Note that if $\mathcal{A}$ is a deterministic algorithm the second term on the RHS of Eq. (53) is zero.

Next we provide an upper bound for $\mathrm{H}(L_1|\tilde{Z},\pi)$. Note that $L_1$ can take values in $\{[1,0]^\intercal,[0,0]^\intercal,[0,1]^\intercal\}$ as it is assumed that $\mathcal{A}_n$ is consistent and, by construction, at each column of $\tilde{Z}^\pi$ one of the points is selected for the training set. Due to the symmetry imposed by $\pi$ and $U$ we have with probability one

$$\mathbb{P}^{\tilde{Z}}[L_1 = [1,0]^\intercal] = \mathbb{P}^{\tilde{Z}}[L_1 = [0,1]^\intercal]. \tag{54}$$

Then, note that there exists a function $\kappa$, depending only on $\mathcal{A}_n$, such that

$$\kappa(\tilde{Z}) = \mathbb{P}^{\tilde{Z}}[L_{\bar{U}_1,1} = 1]. \tag{55}$$

We claim that the common value in Eq. (54) is given by $\frac{\kappa(\tilde{Z})}{2}$. This claim can be easily proved by taking the expectation with respect to $\bar{U}_1$ in Eq. (54) and Eq. (55).

Then, we will show that using Lemma H.3, $\kappa(\tilde{Z})$ is given by the average leave-one-out error of $\mathcal{A}$ over $\tilde{Z}$. Given a training set $S = ((x_1,y_1),\ldots,(x_n,y_n)) \in \mathcal{Z}^n$ and a test point $z = (x,y) \in \mathcal{Z}$, let $P_e(S;z)$ denote the probability that $\mathcal{A}_n$ makes a mistake in predicting the label $x$. Using this notation, we have

$$\kappa(\tilde{Z}) = \mathbb{E}^{\tilde{Z}}\left[P_e((\tilde{Z}^{\pi}_{U_1,1},\ldots,\tilde{Z}^{\pi}_{U_n,n});\tilde{Z}^{\pi}_{\bar{U}_1,1})\right] \tag{56}$$

$$= \mathbb{E}_{J \sim \mathrm{Unif}([2n]_{n+1})}\frac{1}{n+1}\sum_{j\in J} P_e(\tilde{Z}_{J-\{j\}};\tilde{Z}_j), \tag{57}$$

where in the last step we have used Lemma H.3. Then, by the fact that conditioning reduces the entropy we have $\mathrm{H}(L_1|\pi,\tilde{Z}) \leq \mathrm{H}(L_1|\tilde{Z})$. As shown above,

$$\mathbb{P}^{\tilde{Z}}[L_1 = [1,0]^\intercal] = \mathbb{P}^{\tilde{Z}}[L_1 = [0,1]^\intercal]$$

$$= \frac{1}{2}\kappa(\tilde{Z})$$

$$= \frac{1}{2}\mathbb{E}_{J \sim \mathrm{Unif}([2n]_{n+1})}\frac{1}{n+1}\sum_{j\in J} P_e(\tilde{Z}_{J-\{j\}};\tilde{Z}_j). \tag{58}$$

Then, by the fact that the $L_1$ can take values in $\{[1,0]^\intercal, [0,0]^\intercal, [0,1]^\intercal\}$ and Eq. (58) we obtain

$$
\begin{aligned}
\mathrm{H}(L_1|\tilde{Z}) &= \mathbb{E}[-(1-\kappa(\tilde{Z}))\log(1-\kappa(\tilde{Z})) - \kappa(\tilde{Z})/2\log(\kappa(\tilde{Z})/2) - \kappa(\tilde{Z})/2\log(\kappa(\tilde{Z})/2)] \\
&= \mathbb{E}[\mathrm{H}_b(\kappa(\tilde{Z})) + \kappa(\tilde{Z})\log(2)], \quad\quad\quad (59)
\end{aligned}
$$

where $\mathrm{H}_b(\cdot)$ denotes the binary entropy function.

Finally, we find a closed form expression for $\mathrm{H}(L_1|\tilde{Z}, \pi, U)$. Since it is assumed that $\mathcal{A}_n$ is consistent we have $L_{U_i,i} = 0$ a.s.. Therefore $\mathrm{H}(L_1|\tilde{Z}, \pi, U) = \mathrm{H}(L_{\bar{U}_1,1}|\tilde{Z}, \pi, U)$. Then, we can write

$$
\begin{aligned}
\mathrm{H}(L_{\bar{U}_1,1}|\tilde{Z}, \pi, U) &= \mathbb{E}[\mathrm{H}^{\tilde{Z},\pi,U}(L_{\bar{U}_1,1})] \\
&= \mathbb{E}\big[\mathbb{E}^{\tilde{Z}}[\mathrm{H}^{\tilde{Z},\pi,U}(L_{\bar{U}_1,1})]\big] \\
&= \mathbb{E}\big[\mathbb{E}^{\tilde{Z}}\mathrm{H}_b([P_e((\tilde{Z}^\pi_{U_1,1}, \ldots, \tilde{Z}^\pi_{U_n,n}); \tilde{Z}^\pi_{\bar{U}_1,1}))]\big] \\
&= \mathbb{E}\big[\mathbb{E}_{J \sim \mathrm{Unif}([2n]_{n+1})}\frac{1}{n+1}\sum_{j\in J}\mathrm{H}_b(P_e(\tilde{Z}_{J-\{j\}}; \tilde{Z}_j))\big], \quad\quad (60)
\end{aligned}
$$

where in the last line we have used Lemma H.3. Finally by combining Eq. (59), Eq. (60), and Eq. (53) we obtain the stated result in Eq. (38).

# I  Description of the One-Inclusion Graph Prediction Algorithm

In this part we provide a short description of the one-inclusion transductive learning algorithm of Haussler et al. [5]. Let $\bar{X} = (x_1, \ldots, x_n) \in \mathcal{X}^n$ and let $\mathcal{H}$ be a concept class with VC dimension $d$. Let $\mathcal{H}_{|\bar{X}}$ be the equivalence class induced by $\mathcal{H}$ on the instances given in $\bar{X}$. Similarly for a classifier $h \in \mathcal{H}$, we can define $h_{|\bar{X}}$ as the restriction of $h$ to the instances in $\bar{X}$. For every $h \in \mathcal{H}_{|\bar{X}}$, let $v_h = (h(x_1), \ldots, h(x_n)) \in \{0,1\}^n$. Haussler et al. [5] defined the one-inclusion graph of $\bar{X}$ denoted by $\mathcal{G}_\mathcal{H}(\bar{X}) = (V, E)$ as follows. $\mathcal{G}_\mathcal{H}(\bar{X})$ has the vertex set $V = \{v_h : h \in \mathcal{H}_{|\bar{X}}\}$, and $(v_h, v_{h'}) \in E$ if and only if the hamming distance of $v_h$ and $v_{h'}$ is one. For an example of $\mathcal{G}_\mathcal{H}(\bar{X})$ for the concept class of intervals in one dimension, see Fig. 1 of [5]. Consider *probability assignment mapping* $f_{\mathcal{G}_\mathcal{H}(\bar{X})} : E \times V \to [0,1]$ such that for each edge $e$ incident to $v_h$ and $v_{h'}$ the following two conditions holds.

1.  for all $h'' \in \mathcal{H}_{|\bar{X}}$ with $h'' \neq h$ and $h'' \neq h'$, we have $f_{\mathcal{G}_\mathcal{H}(\bar{X})}(e, v_{h''}) = 0$.
2.  $f_{\mathcal{G}_\mathcal{H}(\bar{X})}(e, v_h) \geq 0$, $f(e, v_{h'}) \geq 0$, and $f_{\mathcal{G}_\mathcal{H}(\bar{X})}(e, v_h) + f_{\mathcal{G}_\mathcal{H}(\bar{X})}(e, v_{h'}) = 1$.

For every $i \in [n]$ and $h \in \mathcal{H}_{|\bar{X}}$, let $c_{i,h} \subset \mathcal{H}_{|\bar{X}}$ be the set of all the hypotheses in $\mathcal{H}_{|\bar{X}}$ whose restriction to $\bar{X} \setminus \{x_i\}$ equals to $h_{|\bar{X}\setminus\{x_i\}}$. Let $h^\star \in \mathcal{H}$, consider a realizable $S = ((x_1, y_1), \ldots, (x_n, y_n))$ where $y_i = h^\star(x_i)$ for $i \in [n]$. Assume $S_{n-1} = ((x_1, y_1), \ldots, (x_{n-1}, y_{n-1}))$ is given to a learner and the learner aims to predict the label of $x_n$. It is immediate to see that the set of hypotheses consistent with $S_{n-1}$ is $c_{n,h^\star}$. Clearly, $|c_{n,h^\star}| \in \{1, 2\}$ and if $|c_{n,h^\star}| = 1$, we know that the target hypothesis is $h^\star$. But, what should the learner do when $|c_{n,h^\star}| = 2$?

Using $\mathcal{G}_\mathcal{H}(\bar{X})$ we can think of the case $|c_{n,h^\star}| = 2$ as there is a vertex $v_{h'}$ adjacent to $v_{h^\star}$, and $v_{h'}$ and $v_{h^\star}$ differ in $n$-th position. Assume $|c_{n,h^\star}| = 2$ and $e_{n,h^\star} = \{h^\star, h'\}$. Using the probability assignment mapping $f_{\mathcal{G}_\mathcal{H}(\bar{X})}$ of $\mathcal{G}_\mathcal{H}(\bar{X})$, the strategy proposed by Haussler et al. [5] predicts the label $x_n$ to be $h'(x_n)$ with probability $f_{\mathcal{G}_\mathcal{H}(\bar{X})}(e, v_{h^\star})$, and to be $h^\star(x_n)$ with probability $f_{\mathcal{G}_\mathcal{H}(\bar{X})}(e, v_{h'})$. By identifying a deep combinatorial property of $\mathcal{G}_\mathcal{H}(\bar{X})$, Thm. 2.3 in [5] shows that there exists a probability assignment mapping $f_{\mathcal{G}_\mathcal{H}(\bar{X})}$ with the mentioned properties such that $\sum_{e\in E} f_{\mathcal{G}_\mathcal{H}(\bar{X})}(e, v_h) \leq d$ for all $h \in \mathcal{G}_\mathcal{H}(\bar{X})$, and finding such a mapping is computationally easy. Moreover, Haussler et al. [5] show there exists *deterministic* probability assignment for every one-inclusion graph such that $f_{\mathcal{G}_\mathcal{H}(\bar{X})}(e, v_h) \in \{0, 1\}$ for all $e \in E$ and $v_h \in V$, and $\sum_{e\in E} f_{\mathcal{G}_\mathcal{H}(\bar{X})}(e, v_h) \leq d$ for all $h \in \mathcal{G}_\mathcal{H}(\bar{X})$.