# OpenReview forum: "Towards a Unified Information-Theoretic Framework for Generalization"
_NeurIPS.cc/2021/Conference — NeurIPS 2021 Spotlight_

### Official Review · Reviewer_6q6T · 2021-07-12

**Rating:** 8
**Confidence:** 3

**Summary:**

This work investigates the expressiveness of the CMI framework introduced in [1]. The main contributions of this paper are:

1. The authors show that an empirical variant of CMI can be easily used to express bounds every improper learning.
2. The authors apply their general CMI bound on SVM, and improve previous results by a logarithmic factor.
3. VC classes for which proper learning cannot be bounded by any function of the VC dimension are considered
4. They prove a general reduction showing that the leave-one-out is expressible via the CMI framework.

**Ethical Concerns:**

No.

**Limitations And Societal Impact:**

Yes.

**Main Review:**

That paper is well-written, motivated, and significantly improve our understanding of the CMI framework.
The results are quite interesting, and I found the proofs of Theorems 3.4 and 5.4 beautiful and non-trivial.
To me, this is a solid contribution, and I do not have any interesting comments/concerns.

**Time Spent Reviewing:**

48

---

> ### Author Response · Authors · 2021-08-10
> **Response**
>
> We thank this reviewer for all the positive comments!
>
> We are glad that you find our paper **well-written** with **solid contributions**. We hope the reviewer has enjoyed reading our paper.

---

### Official Review · Reviewer_dG22 · 2021-07-15

**Rating:** 6
**Confidence:** 4

**Summary:**

The paper discusses using the Conditional Mutual Information (CMI) framework to express bounds for generalization error. The authors discuss the use of CMI for several settings. In particular, they (i) show that their main result yields a tight bound for SVMs, (ii) answer a conjecture in the negative, and (iii) show that their result yields near-optimal bounds on algorithms with a "leave-one-out error guarantee".

**Limitations And Societal Impact:**

No discussion on societal impact is available.

**Main Review:**

1. Originality

The authors compare adequately with the work of Steinke and Zakynthinou, and provide some references to other work.

2. Quality

The claims are purely theoretical. A minor point:

a. In line 190, the authors say that "the bound in Theorem 3.2 cannot be improved from O(k log n) to O(k)... so the bound in Theorem 3.2 is optimal". This is not logically sound,  since there are other orders between k and k log n (such as k log (log n))---is there a proof that  O(k log n) is the "best possible"?

3. Clarity

The paper is not written very clearly. For example, in Theorem 1, the basic terms n is not defined. The same issue also occurs in the first paragraph, where the IOMI is denoted by "IOMI_D(A_n)" but both D and n are not defined.  Although the definitions come later on in the text, it might be better to have definitions appear earlier.

The wording is unclear in many places, and I think it would be helpful to add clear definitions and/or descriptions to make reading easier:

a. In Line 39 what does "class of one-dimensional thresholds over {1,...,m}" mean?

b. In Line 81 what does "release of the set of all consistent classifiers" mean?

c. The definitions within the paragraph at Line 128 along with footnote 2 can perhaps be thinned down a bit.

4. Significance

Understanding guarantees of algorithms helps both researchers and practitioners gain confidence of using these algorithms, and therefore from this perspective, the results of this paper may be important. It is however unclear from this paper the significance of looking at CMI compared to other measures such as the usual risk. It might be helpful if some of the proofs are moved to the appendix and the space is used to address the importance of these results. Finally, I believe that improving clarity within later revisions would also help to clarify significance and importance.

5. Additional comments

Some typos:

a. "in realizable" in Line 3 doesn't make sense.

b. In line 127, is the definition of KL divergence missing an integral somewhere?

c. In line 253 "showing" is probably "show".

d. In line 310 "the sample the supersample" may be a typo.

6. Update after reading rebuttal and other reviews: The authors have addressed my concerns and I am adjusting my scores.

**Time Spent Reviewing:**

3 hours

---

> ### Author Response · Authors · 2021-08-10
> **Response**
>
> # Summary
> We thank the reviewer for reviewing our paper. Our understanding is the main criticism of the reviewer is the significance of looking at CMI compared to other measures such as the usual risk. Also, the reviewer questions the significance and importance of the results in the paper. We start our rebuttal by addressing these questions.
>
> # The significance of considering the CMI framework
>
> There are various generalization frameworks exists in the literature. However, these different methods for proving generalization guarantees are largely disconnected from one another, and, it is, in general, not possible to compare or combine techniques. As an example, consider the hypothesis class of thresholds over real line. This class under VC framework has sharp generalization guarantees (VC dimension is one). However, PAC-Bayes [LM20], input-output mutual information [NSY18], and privacy [BNS19] fail to explain the generalization of this simple hypothesis class. This is a very active line of research to understand the limitations of different frameworks and find an unifying framework which is compatible with the existing frameworks. The CMI introduced by Steinke and Zakynthinou (2020) is a new framework which has potential to provide a unifying framework or language to study generalization. In particular, Steinke and Zakynthinou in their main paper show that CMI can tie together existing frameworks, including VC, sample compression, privacy, input-output mutual information, and uniform stability. Therefore, in this work, we investigate further the expressiveness of the CMI framework of and the prospect of using it to provide a unified framework for proving generalization bounds in the realizable setting.
>
> To address the significance of the results in the current paper, note the results proved by Steinke and Zakynthinou for VC and sample compression schemes are suboptimal in some cases. The current paper aims to answer the question of whether the CMI framework can be used to obtain optimal bounds for VC and sample compression frameworks. All of our results in this paper are positive, and we found that CMI framework can be connected to VC and sample compression frameworks to find *optimal* bounds in various cases.
>
> As per your suggestion, we removed the proof sketch of Theorem 4.8 to incorporate your feedback.
> We will gladly add further discussion on the significance of our results in the introduction in the camera-ready version as an extra page available. Also, we will include more explanation  to improve approachability of the paper.
>
>
> # Significance of considering a generalization framework compared to the usual risk
>
> Evaluating "directly" the risk and using a "generalization framework" have their own advantages and disadvantages. While directly computing the risk gives a precise answer to a particular problem, this approach gives little understanding of the common properties among different learning problems which ensures generalization. On the other hand, the generalization frameworks let us take a principled approach in understanding why a learning algorithm generalizes from the training data to the test data. In summary, there is a trade-off between "precision" and "generality".
>
>
>
>
> # Answers to Specific Questions
>
>
> *Q1: In line 190, the authors say that "the bound in Theorem 3.2 cannot be improved from $O(k log n)$ to $O(k)$... so the bound in Theorem 3.2 is optimal". This is not logically sound, since there are other orders between $k$ and $k \log n$ (such as $k \log (\log n))$---is there a proof that $O(k \log n)$ is the "best possible"?*
>
> A1: In fact there is a proof which shows that $O(k\log(n))$ is the best achievable CMI if we consider general sample compression schemes. The proof comes from the fact that there exist compression schemes of size $k$ for which, for
> certain distributions, one can prove lower bound  $\mathbb{E}[R_{\mathcal{D}}(\mathcal{A}_n)]=\Omega({k \log (n)}/{n})$  where  $\mathcal{A}_n (\cdot)=\rho(\kappa(\cdot))$
>  [FW94, Sec. 5].  Also, from Eq.(2), we know that for consistent classifiers we have $ \mathbb{E}[R_\mathcal{D}(\mathcal{A}_n)] \leq  O( \text{CMI}_\mathcal{D} (\mathcal{A}_n)/n) $. Therefore, by combining these two equations the stated optimally result follows. We have revised the text and made the proof completely clear.
>
>
>
>
> *Q2: The paper is not written very clearly. For example, in Theorem 1, the basic terms $n$ are not defined. The same issue also occurs in the first paragraph, where the IOMI is denoted by $\text{IOMI}_{\mathcal{D}}(\mathcal{A}_n)$ but both $\mathcal{D}$ and $n$ are not defined. Although the definitions come later on in the text, it might be better to have definitions appear earlier.*
>
> A2:Thanks for pointing it out to us.
> We have revised the beginning of the introduction and first we carefully define $\mathcal{D}$ and $n$ before introducing input--output mutual information.
>
>
>
> *Q3: In Line 39 what does "class of one-dimensional thresholds over $\set{1,...,m}$" mean?*
>
> A3:	 We have added the definition of this concept class in the introduction which reads as follows:
> 	 ``
> 	 Let the input space be $\mathcal{X}=\set{1,...,m}$. 	Let $k \in \mathbb{N}$ and $h_k: \mathcal{X} \to \set{0,1}$ define as $h_k(x)=1[x> k]$. Then, the class of one-dimensional thresholds over $\set{1,...,m}$ is $\mathcal{H}_m=\set{h_k | k\in \mathbb{N}}$.
>
>
> *Q4: In Line 81 what does "release of the set of all consistent classifiers" mean?*
>
>
> A4:  Consider a scenario where there is a data holder with access to the training data. After observing the training samples $S_n$, the data holder constructs and **releases** the set of *all* classifiers in $\mathcal{H}$  achieving zero training error on $S_n$, i.e., $V_\mathcal{H} [S_n]=\set{h \in \mathcal{H} | \hat{R}_{S_n}(h)=0}$.
>
> Then, there is a learner that only knows the set $V_{\mathcal{H}}[S_n]$, and it wishes to select an element $\hat{h}$ from $V_{\mathcal{H}}[S_n]$.  In summary, we used the term of releasing the set of all consistent classifiers  to model the above-mentioned scenario where we have two parties, i.e., data holder and learner.
>
>
> *Q5: The definitions within the paragraph at Line 128 along with footnote 2 can perhaps be thinned down a bit.*
>
> A5: We  followed your suggestion and moved the first footnote to the main body since it gives an intuition behind the definition of CMI. We also removed the second footnote and instead provided a reference to a standard probability theory textbook.
>
> *Q6: Typos.*
>
>
> A6: We fixed all the typos. Regarding the definition of KL divergence, in Line 127 we introduced a notation for the expectation operation which is $\smash{P[f] = \int f d P}$. Using this notation we believe that the definition of KL in its current format is correct.
>
>
>
> # References:
>
>
> [BNS19] Beimel, A., Nissim, K.,  Stemmer, U. (2019). Characterizing the Sample Complexity of Pure Private Learners. J. Mach. Learn. Res., 20, 146-1.
>
> [NSY18] Nachum, I., Shafer, J.,  Yehudayoff, A. (2018). A direct sum result for the information complexity of learning. In Conference On Learning Theory (pp. 1547-1568). PMLR.
>
> [LM20] Livni, R.,  Moran, S. (2020). A limitation of the pac-bayes framework. Advances in Neural Information Processing Systems, 33.
>
> [FW94] S. Floyd and M. Warmuth. “Sample compression, learnability, and the Vapnik-Chervonenkis dimension”.Machine learning 21.3 (1995), pp. 269–304.

---

> > ### Author Response · Authors · 2021-08-24
> > **Thanks for reading our response.**
> >
> > Dear Reviewer,
> >
> > We thank the reviewer for reading our response. Are there any outstanding issues that our response has not addressed or only partially addressed? We'd welcome guidance on how we might improve the paper to the point where it would earn an unqualified "Accept" from you.
> >
> > Thanks.

---

### Official Review · Reviewer_EVBd · 2021-07-17

**Rating:** 9
**Confidence:** 4

**Summary:**

The paper studies the ability of the recently proposed conditional mutual information framework (CMI) to characterize distribution-free generalization bounds of the expected excess risk for learning algorithms in the classification realizable setting of VC classes.
They first show that CMI can be used to yield rates (sub-optimal in some cases) for any improper learning algorithm whose output hypotheses belong to a VC class. Next, they show that the CMI framework yields optimal bounds on the expected excess risk for stable sample compression schemes. An application shows that CMI yields optimal rates for the SVM algorithm. As for proper learning, it is shown that in some cases the CMIF cannot yield optimal rates by showing that the CMI cannot be bounded in terms of the VC-dimension alone. They also show that a finite star number (a recently introduced combinatorial property of VC classes) implies a finite CMI. Lastly, for non-VC classes, they show that any learning algorithm with a “leave-one-out” bound of order $O(1/n)$ yields an (empirical) CMI bound of order $O(log n)$. As an application, they obtain near-optimal bounds for the one-inclusion graph algorithm of Haussler et al. [5] for improper learning of VC classes and give an example where the rates are optimal. Open questions regarding the characterization are also discussed.

**Ethical Concerns:**

None.

**Limitations And Societal Impact:**

Yes.

**Main Review:**

Overall, the paper is well written with novel, interesting and important results. The results seem technically correct, but I didn't verify all the proofs. I only have some minor comments on the presentation.

- To make the context clearer, I'm missing a clear statement in the introduction of what are the achievable optimal rates in the realizable setting for proper and improper learning, and more generally, when rates of O(d/n) and O(dlogn/n) are optimal. I hope the authors can refine their discussion in this regard in lines 56-66 (in particular, the phrase "some cases" in line 61 is unclear) and in the contribution list. A discussion on whether the compression framework characterizes optimal rates will also be beneficial (but might be considered as out of scope).

- At the end, I could not readily say when the CMI framework might fail to characterize optimal rates in the proper/improper learning cases. For example, it is known there are cases where no proper learning algorithm can achieve the optimal rate O(d/n). According to item 3 in Contributions, CMI does not yield O(d/n) for proper learning in some cases as well. So how are these two cases related?

- Please give more details in the Statement of Theorem 4.4, indicating clearly in what way Statement 1 fails. I believe that a clear discussion on the implications of this Theorem for the ability of the CMI framework to characterize optimal rates for proper learning would be beneficial to the reader.

**Time Spent Reviewing:**

9

---

> ### Author Response · Authors · 2021-08-10
> **Response**
>
>
> We thank the reviewer for the positive review as well as the suggestions for improvement.
>
> # Answers to the questions:
>
> *Q1:To make the context clearer, I'm missing a clear statement in the introduction of what are the achievable optimal rates in the realizable setting for proper and improper learning, and more generally, when rates of O(d/n) and O(dlogn/n) are optimal. I hope the authors can refine their discussion in this regard in lines 56-66 (in particular, the phrase "some cases' ' in line 61 is unclear) and in the contribution list. A discussion on whether the compression framework characterizes optimal rates will also be beneficial (but might be considered as out of scope).*
>
>
> A1: We agree this would be useful to readers. We can describe the landscape and then relate it to our results, also pointing to the open questions we raise.
>
> One key distinction is, of course, proper versus improper learning: For every VC class with dimension $d$, there exists a consistent and improper learning algorithm that achieves $O(d/n)$ risk under realizability, and this bound is optimal [H16b, HLW94]. The situation for proper learning is much more complicated. In general, the achievable rate for the proper learning of VC classes is off by a log factor, i.e., $O(d\log(n)/n)$. [BHMZ20] showed that when the dual Helly and hollow-star number agree, which are combinatorial complexity measures of the class, then they characterize the existence of an optimal proper learner. Also, a subclass of VC for which the log factor provably cannot be removed using proper learners is characterized in [BHMZ20, Thm.11].  Moreover, for general ERMs, Hanneke [H16a] shows the finiteness of the star number is a necessary and sufficient condition under which we can remove the log factor using any ERMs.
>
> It is interesting to note that the results of [MY16] showed a connection between the general sample compression schemes and VC classes. However, it is not known whether, in general, the optimal rates for VC classes are always witnessed by compression schemes.
>
> We will add an overview of the existing optimal rates known for VC classes in the introduction in the camera ready as an extra page available.
>
>
> *Q2: At the end, I could not readily say when the CMI framework might fail to characterize optimal rates in the proper/improper learning cases. For example, it is known there are cases where no proper learning algorithm can achieve the optimal rate O(d/n). According to item 3 in Contributions, CMI does not yield O(d/n) for proper learning in some cases as well. So how are these two cases related?*
>
>
>
> A2: Note the results of our paper do not present any limitations of CMI for both sample compression schemes and VC classes. In particular, we investigated proper learning of VC classes with finite star number, and proved that we can remove the log factor from CMI to achieve expected risk $O(d/n)$. The results of [H16a] from VC theory show that this class admits proper learners with rate $O(d/n)$, and we recover it using CMI framework. In Section 6 of the paper, we provided several open questions. In the revision, we have expanded and added more details into the open questions and discussed how negative answers to these questions can imply limitations of the CMI framework.
>
> Q3: Please give more details in the Statement of Theorem 4.4, indicating clearly in what way Statement 1 fails. I believe that a clear discussion on the implications of this Theorem for the ability of the CMI framework to characterize optimal rates for proper learning would be beneficial to the reader.
>
> A3: We agree that more discussion would make the interpretation of this result easier. We do not interpret this result as a failure of CMI. In fact,  Theorem 4.4 implies that Statement 1 contradicts with the known lower bounds for VC classes in [BHMZ20, Thm.11].
> In particular, an implication of our result in Theorem 4.4 is that we should focus on the subclasses of VC classes for which there is no lower bound indicating the best achievable rate is $O(d\log(n)/n)$. This is the reason in Section 4.2 we consider a subclass of VC classes with finite star number. This subclass is among the sub-classes that is known we can remove the log factor from the results in the VC theory, and in particular we showed in Theorem 4.9 that we can remove the log factor using CMI as well.
>
> References:
>
> [H16a] S. Hanneke. "Refined error bounds for several learning algorithms". The Journal of Machine Learning Research (2016), pp. 4667–4721
>
>   [H16b] S. Hanneke. "The optimal sample complexity of PAC learning." The Journal of Machine Learning Research (2016), pp. 1319-1333.
>
>  [BHMZ20]  O. Bousquet, S. Hanneke, S. Moran, and N. Zhivotovskiy. “Proper Learning, Helly Number, and an Optimal SVM Bound”. In:Proceedings of the 33rd Conference On Learning Theory. Ed. by J. Abernethy and S. Agarwal. Vol. 125. Proceedings of Machine Learning Research. PMLR, July 2020, pp. 582–609.
>
>  [HLW94] D. Haussler, N. Littlestone, and M. K. Warmuth. “Predicting  $\set{0,1}$-functions on randomly drawn points”. Information and Computation 115.2 (1994), pp. 248–292.
>
> [MY16] Moran, S.,   Yehudayoff, A. (2016). Sample compression schemes for VC classes. Journal of the ACM (JACM), 63(3), 1-10.

---

### Decision · Program_Chairs · 2021-09-27

**Decision:**

Accept (Spotlight)

**Comment:**

The paper makes progress in a new approach for proving generalization bounds.